# MetricEmbedding: Accelerate Metric Nearness by Tropical Inner Product

**Muyang Cao** [1 2]   **Jiajun Yu** [1 2]   **Xin Du** [3]   **Gang Pan** [1]   **Wei Wang** [4 5]

## Abstract

The Metric Nearness Problem involves restoring a non-metric matrix to its closest metric-compliant form, addressing issues such as noise, missing values, and data inconsistencies. Ensuring metric properties, particularly the $O(N^3)$ triangle inequality constraints, presents significant computational challenges, especially in large-scale scenarios where traditional methods suffer from high time and space complexity. We propose a novel solution based on the tropical inner product (max-plus operation), which we prove satisfies the triangle inequality for non-negative real matrices. By transforming the problem into a continuous optimization task, our method directly minimizes the distance to the target matrix. This approach not only restores metric properties but also generates metric-preserving embeddings, enabling real-time updates and reducing computational and storage overhead for downstream tasks. Experimental results demonstrate that our method achieves up to 60× speed improvements over state-of-the-art approaches, and efficiently scales from $1e4 * 1e4$ to $1e5 * 1e5$ matrices with significantly lower memory usage.

## 1. Introduction

Distance matrices are fundamental in various machine learning and optimization tasks, including clustering, K-nearest neighbors (KNN), and recommender systems (Jain et al., 1999; Cover & Hart, 2006; Peterson, 2009; Lü et al., 2012; Piao & Breslin, 2016). Among them, commonly used matrices such as shortest path distance matrices and Euclidean distance matrices must adhere to **metric properties**—specifically, non-negativity, symmetry, and the triangle inequality (Randic et al., 1994). The triangle inequality ensures that for any three points, the direct distance between two points does not exceed the sum of the distances through an intermediary point (Sra et al., 2004; Brickell et al., 2008). These properties are crucial for maintaining the accuracy and stability of downstream applications (Roth et al., 2002; Elkan, 2003; Hussain, 2018).

However, real-world distance matrices often deviate from ideal metric properties due to noise, measurement errors, missing data, and structural inconsistencies. Such deviations can disrupt downstream tasks, leading to unreliable predictions and degraded performance (Indyk, 1999; Russell & Norvig, 2016; Yu et al., 2023). For example, in clustering, an algorithm typically assigns an object to its nearest cluster centroid based on a given distance function (Hartigan et al., 1979; Jain & Dubes, 1988; Park & Jun, 2009). If the distance matrix violates the triangle inequality, objects assigned to the same cluster may appear far apart, leading to unstable clustering. Similar issues arise in shortest-path computations, and recommender systems, where violations of metric properties cause inconsistencies in similarity and ranking.

To address these inconsistencies, the Metric Nearness Problem (MNP) seeks to transform a given distance matrix into the closest metric-compliant matrix while preserving as much original information as possible (Brickell et al., 2008). The primary challenge lies in satisfying the $O(N^3)$ triangle inequality constraints while maintaining computational efficiency. Traditional approaches rely on iterative projection methods, such as the Bregman projection algorithm (Brickell et al., 2008; Sonthalia & Gilbert, 2022) and the HLWB projection algorithm (Li et al., 2023), or learning-based methods that approximate a valid distance metric. However, these methods face high computational cost, poor scalability, and limited real-time adaptability, which hinder large-scale applications needing efficient metric corrections.

In this paper, we introduce tropical inner products into Euclidean space by defining a novel tropical inner product between matrices (or vectors). This construction provides a theoretical foundation for linking tropical algebra to metric matrices, filling a key gap in the literature. Most importantly, we prove that the tropical inner product of two non-negative

[1]College of Computer Science and Technology, Zhejiang University [2]Shanghai Innovation Institute [3]Shool of Software Technology, Zhejiang University [4] The Hong Kong University of Science and Technology (Guangzhou) [5] The Hong Kong University of Science and Technology. Correspondence to: Xin Du <xindu@zju.edu.cn>, Wei Wang <weiwcs@ust.hk>.

*Proceedings of the 42^{nd} International Conference on Machine Learning*, Vancouver, Canada. PMLR 267, 2025. Copyright 2025 by the author(s).

matrices satisfies the triangle inequality, ensuring that it can be used to construct valid metric matrices.

Building on this, we establish the Tropical Algebra-metric representation theorem(TA-metric representation theorem), which reveals the representational space of the tropical inner product of non-negative matrices and its relationship with metric matrices. This result formally characterizes the representational capacity of tropical inner products in metric space. Additionally, we derive a series of theoretical insights into the relationship between tropical inner products and metric matrices, including the relationship between shortest-path distances and tropical inner product.

Leveraging this theoretical framework, we propose MetricEmbedding, a novel approach based on tropical inner products and gradient descent for solving the Metric Nearness Problem (MNP). Specifically, we designed a MetricEmbedding architecture based solely on the max-TA inner product, with each network's parameters aimed at approximating a target matrix. This distinguishes it from traditional MLP(Cybenko, 1989) and other similar deep neural network architectures, including the min-max-plus network(Luo & Fan, 2021).Instead of learning latent distributions, our model directly approximates a target metric matrix using parameterized tropical algebra operations. Furthermore, we enhance deep learning architectures by integrating ReLU activation and tropical inner products, ensuring that the output satisfies the triangle inequality. This enables applications in metric learning and contrastive learning.

Our approach also introduces key improvements in computational efficiency and scalability. By modifying the loss function, our method flexibly adapts to different problem formulations, supports mini-batch optimization, and scales efficiently to large datasets. Additionally, we develop an online prediction mechanism, making our model suitable for real-time applications. Experimental results demonstrate that our method achieves up to 60× speed improvements over state-of-the-art approaches, while efficiently scaling from $10^4 \times 10^4$ to $10^5 \times 10^5$ matrices with significantly lower memory usage.

The contributions of this paper can be summarized as follows:

1. We introduce the tropical inner product into Euclidean space and establish a theoretical framework that connects tropical inner products and metric matrices. Our key result, the TA-metric representation theorem, reveals the representational space of the tropical inner product of non-negative matrices and its relationship with metric matrices, providing a novel and principled way to construct metric matrices.

2. Based on this theoretical findings, we propose Met-

ricEmbedding, a network structure built with tropical inner product, capable of predicting a metric matrix that is sufficiently close to a target distance matrix. It has advantages in scalability, real-time prediction, and computational efficiency.

3. We propose a method based on the tropical inner product, where no changes are needed in the network structure except for adding a layer to guarantee that the network output satisfies the triangle inequality or metric properties. This approach holds significant potential for metric learning and contrastive learning tasks.

## 2. Preliminary

**Tropical Inner Product and Matrix Multiplication**

Tropical algebra replaces traditional operations with tropical ones (Maslov, 1985; Cuninghame-Green, 2012). In the Max-Tropical Semiring, addition is the maximum ($\oplus_{\max}$) and multiplication is standard addition ($\otimes$) (Maragos et al., 2021). The Min-Tropical Semiring is its dual, with addition as the minimum ($\oplus_{\min}$) and multiplication remaining standard addition ($\otimes$) (Joswig, 2021).

Formally, the Max-Tropical Semiring is defined as ($\mathbb{R} \cup \{-\infty\}, \oplus_{\max}, \otimes$), where $\oplus_{\max}$ denotes the maximum operation and $\otimes$ is standard addition, while the Min-Tropical Semiring is defined as ($\mathbb{R} \cup \{\infty\}, \oplus_{\min}, \otimes$), where $\oplus_{\min}$ denotes the minimum operation.

Here,We adapt tropical operations to define the tropical inner product in Euclidean space. Let $\mathbb{R}^n$ denote the Euclidean space of dimension $n$. For two vectors:

$$u = (u_1, u_2, \ldots, u_n) \quad \text{and} \quad v = (v_1, v_2, \ldots, v_n) \in \mathbb{R}^n,$$

the tropical inner product is defined as:

$$\langle u, v \rangle_{\oplus_{\max}} = \max_{1 \leq i \leq n} (u_i + v_i),$$

where $\oplus_{\max}$ denotes tropical addition (maximum) and $\otimes$ denotes standard addition. This product computes the maximum sum of corresponding elements from the two vectors.

This concept extends naturally to matrices. Given two matrices $A \in \mathbb{R}^{n \times m}$ and $B \in \mathbb{R}^{m \times p}$, the tropical matrix inner product (or tropical matrix multiplication) is defined as:

$$C = A \odot_{\max} B \in \mathbb{R}^{n \times p}$$

where $\odot_{\max}$ denotes the tropical multiplication using the maximum operation. Each element $C(i, j)$ of the resulting matrix $C$ is computed using tropical operations as follows:

$$C_{i,j} = \max_{1 \leq k \leq m} (A_{i,k} + B_{k,j}).$$

Alternatively, the tropical matrix product can be defined

using the minimum operation, denoted $\odot_{\min}$, where:

$$C_{i,j} = \min_{1 \leq k \leq m} (A_{i,k} + B_{k,j}).$$

As shown in **Appendix A**, in both cases, tropical matrix multiplication computes the maximum or minimum "path" sum from row $i$ of matrix $A$ to column $j$ of matrix $B$, considering all possible intermediate paths through index $k$.

**Metric Nearness Problem**

A **metric matrix** $M \in \mathbb{R}^{N \times N}$ must satisfy the following conditions for all $i, j, k \in \{1, 2, \ldots, N\}$:**Non-negativity**: $M_{ij} \geq 0$;**Symmetry**: $M_{ij} = M_{ji}$;**Triangle Inequality**: $M_{ij} \leq M_{ik} + M_{kj}$ for all $i, j, k$.The set $M_N$ represents the space of all valid metric matrices of size $N \times N$, where each matrix $M \in M_N$ satisfies the conditions of a metric, i.e., non-negativity, symmetry, and the triangle inequality (Sra et al., 2004; Brickell et al., 2008; Li et al., 2023).

On the other hand, **non-metric matrices** may not satisfy the triangle inequality or symmetry, and they may contain negative values or inconsistent entries, making them unsuitable for tasks that depend on metric structures, such as clustering or optimization (Brickell et al., 2008).

Given a distance matrix $D \in \mathbb{R}^{N \times N}$, the goal of the Metric Nearness problem is to find a metric matrix $M \in M_N$, such that $M$ minimizes the difference from $D$ under some 'closeness' metric, while satisfying the properties of a metric matrix (Sra et al., 2004; Brickell et al., 2008; Sonthalia & Gilbert, 2022; Li et al., 2023). The problem can be formulated as:

$$M = \arg \min_{X \in M_N} \|W \odot (X - D)\|_p$$

where $M_N$ denotes the set of all valid $N \times N$ metric matrices, $W$ is a symmetric weight matrix, indicating the trust level in the entries of $D$, $\odot$ represents element-wise multiplication, $p$ is the norm used to measure the "closeness" of the matrices.

The goal of this optimization problem is to find a metric matrix $M$ that is the closest to $D$ in terms of the weighted $p$-norm difference.

Additionally, let $B \in \mathbb{R}^{N \times N}$ be a square matrix. We define $B^{\text{off}} \in \mathbb{R}^{N \times N}$ as the matrix obtained by setting the diagonal elements of $B$ to zero, i.e., $B^{\text{off}} = B - diag(B)$ where $diag(B)$ represents the diagonal matrix consisting of the diagonal elements of $B$.

## 3. Methods

### 3.1. TA-metric Representation Theorem

In this subsection, we investigate the relationship between tropical operations and metric matrices.The proof and the

other theorems in the **Appendix B**.

**Theorem 1:** If matrix $A \in \mathbb{R}^{N \times N}$ satisfies the triangle inequality, and matrix $B \in \mathbb{R}^{N \times N}$ satisfies the triangle inequality, then the matrix $C = A + B \in \mathbb{R}^{N \times N}$ also satisfies the triangle inequality.

**Theorem 2:** If matrix $A \in \mathbb{R}^{N \times N}$ is a metric matrix, then for any positive constant $\alpha$, $\alpha A$ is also a metric matrix.

**Theorem 3:** If a matrix $A$ satisfies the triangle inequality, then the matrix $A + A^T = B$, and after setting the diagonal elements of $B$ to 0, the resulting matrix satisfies the metric properties.

**Theorem 4:** If $A$ and $B$ are both $N \times N$ metric matrices, then the element-wise maximum of $A$ and $B$, denoted as $\max(A, B)$, is also a metric matrix.

**Theorem 5:** Let $A \in \mathbb{R}^{N \times K}$ and $B \in \mathbb{R}^{K \times N}$ be non-negative matrices, and let $C = A \odot_{\max} B$. Then for any $i, j, k \in \{1, \ldots, N\}, we\, have : C_{ik} + C_{kj} \geq C_{ij}$.

**Proof**: By definition, $C_{ij} = \max_r(A_{ir} + B_{rj})$. Assume that when $r = s$, equality holds, i.e., $C_{ij} = A_{is} + B_{sj}$. $A_{is} \leq A_{is} + B_{sk} \leq C_{ik} = max_r(A_{ir} + B_{rk})$ and $B_{sj} \leq A_{ks} + B_{sj} \leq C_{kj} = max_r(A_{kr} + B_{rj})$, we have:

$$C_{ij} \leq C_{ik} + C_{kj}.$$

ie.the tropical inner product satisfies the triangle inequality.

**Theorem 6:** Let $A$ be a non-negative matrix of size $N \times K$. Define $B = A \odot_{\max} A^T$ as the tropical inner product of $A$ with its transpose. Let $B^{\text{off}}$ be the matrix obtained by setting the diagonal elements of $B$ to zero. Then, $B^{\text{off}}$ is a metric matrix.

**Theorem 7:** Let $A \in \mathbb{R}^{N \times k}$ and $B \in \mathbb{R}^{k \times M}$ be non-negative matrices, where $k > N * M$. Then, there exist matrices $A' \in \mathbb{R}^{N \times k'}$ and $B' \in \mathbb{R}^{k' \times N}$, with $k' \leq N * M$, such that:

$$A' \odot_{\max} B' = A \odot_{\max} B.$$

**Theorem 8: max-TA-metric Representation Theorem**

Let $\Omega$ denote the set of all matrices produced by tropical inner products of non-negative matrices:

$$\Omega = \left\{ \mathbf{M} \,\middle|\, \mathbf{M} = (\mathbf{X} \odot_{max} \mathbf{Y}^T)^{\text{off}}, \, \mathbf{X}, \mathbf{Y} \in \mathbb{R}_{\geq 0}^{N \times K} \right\},$$

where $(\mathbf{X} \odot_{max} \mathbf{Y^T})_{ij} = \max_k(\mathbf{X}_{ik} + \mathbf{Y}_{jk})$.

Let $\Psi \subset \mathbb{R}_{\geq 0}^{N \times N}$ be the set of matrices $\mathbf{D}$ satisfying:

- $\mathbf{D}_{ii} = 0 \quad \forall i \in \{1, 2, ..., N\}$,

- $\mathbf{D}_{ij} \leq \min_{k_1 \neq i} \mathbf{D}_{ik_1} + \min_{k_2 \neq j} \mathbf{D}_{k_2 j} \quad \forall i, j \in \{1, 2, ..., N\}$,

Then, $\Omega = \Psi$. The set of matrices generated via the tropical inner product (with diagonals removed) from non-negative matrices constitutes a specific subset of the set of non-negative matrices with zero diagonals that satisfy the triangle inequality.

Besides any matrix $\mathbf{D} \in \Psi$ also satisfies the standard triangle inequality,

$$\mathbf{D}_{ij} \leq \mathbf{D}_{ik} + \mathbf{D}_{kj} \quad \forall i, j, k \in \{1, 2, ..., N\}$$

as a consequence of the combination of the two conditions above.

### 3.2. Optimization Strategy: MetricEmbedding

In this work, we propose **MetricEmbedding** for solving the Metric Nearness Problem, aiming to optimize the non-negative matrix $A$ such that the tropical inner product $(A \odot_{\max} A^T)^{\text{off}}$ closely approximates the target matrix $D$, while preserving essential metric properties like the triangle inequality, as established in Theorems 5,6. For convenience in discussion, we set the optimization objective as the squared Frobenius norm between $(A \odot_{max} A^T)^{\text{off}}$ and $D$:

$$\min_A \|(A \odot_{max} A^T)^{\text{off}} - D\|_F^2$$

Here, $A \in \mathbb{R}_{\geq 0}^{N \times k}$ is the matrix to optimize, where $N$ is the number of data points and $k$ is a hyperparameter controlling the model's expressiveness. $D \in \mathbb{R}^{N \times N}$ is the target distance matrix.

Inspired by tropical rank (Guillon et al., 2015) and studies on matrix factorization complexity (Shitov, 2014), and based on the analysis of Theorem 7, we observe that increasing the parameter $k$ can indeed enhance the expressive power of the model when constructing tropical algebraic representations. However, the marginal gains diminish as $k$ increases. At the same time, larger model capacity may introduce optimization challenges such as gradient sparsity. Therefore, in practical modeling, a trade-off must be made between expressive power and computational efficiency. According to Theorem 8, although the tropical inner product space is a subset of a metric space, it still possesses sufficient expressive capability to capture essential metric properties. This ensures that the generated matrix retains desirable metric characteristics while maintaining high computational efficiency. Based on this, our model design emphasizes the efficient approximation of the target matrix $D$, while also controlling resource consumption during training and inference.

To further improve the modeling capacity for complex structures, MetricEmbedding draws inspiration from multilayer perceptrons (MLPs) (Cybenko, 1989) and other deep neural network architectures. It adopts a multi-layer structure

based on tropical algebra, progressively building richer representations of the target matrix to overcome the limitations of single-matrix modeling. By using tropical inner products ($\odot_{\max}$) and tropical addition ($\oplus$), MetricEmbedding preserves the metric properties and allows the model to capture more intricate patterns, effectively balancing expressiveness with computational efficiency.

The architecture of MetricEmbedding can be formalized as follows: $\mathbf{O} = ((\cdots((W_0 \odot_{max} W_1) \oplus b_1) \cdots b_{k-1}) \odot_{max} W_k) \oplus b_k$ Where:

- $W_0 \in \mathbb{R}_{\geq 0}^{N \times d_0}$ is the learnable input matrix, where $N$ is the number of samples and $d_0$ is the number of features. Unlike traditional MLPs, where the input is a fixed vector derived from the input data, $W_0$ is treated as a learnable non-negative matrix.

- $\forall i \in \{1, 2, \ldots, k\}, W_i \in \mathbb{R}_{\geq 0}^{d_{i-1} \times d_i}$ are the non-negative weight matrices, connecting layer $i-1$ to layer $i$, ensuring the output matrix remains non-negative.

- $\forall i \in \{1, 2, \ldots, k\}, b_i \in \mathbb{R}_{\geq 0}^{d_i}$ are the non-negative bias terms at each layer, These bias terms serve as optional adjustments to shift the output of each layer while maintaining the local structural properties of the data.

- $\odot_{\max}$ denotes the max-tropical inner product. This operation preserves the local structure of the output at each layer.

- $\oplus$ denotes tropical addition (element-wise maximum operation), similar to the addition operation in MLPs, but replacing addition with the element-wise maximum, ensuring the preservation of the local structural features of the output.

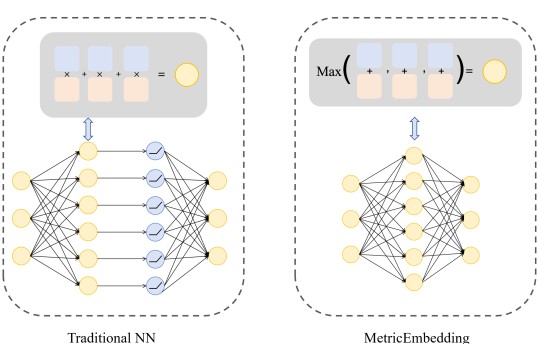

Traditional NN        MetricEmbedding

*Figure 1.* Comparison between MLP and MetricEmbedding. MLP uses ReLU activation and standard matrix multiplication, while MetricEmbedding removes ReLU, replaces matrix multiplication with max-tropical inner product, and modifies the bias addition to max operation.

The training procedure for MetricEmbedding follows the typical neural network optimization pipeline, with some key modifications. The weight matrices $W_0, W_1, \ldots, W_d$ and

---

**Algorithm 1** Training Procedure for MetricEmbedding

---

**Input:** Target matrix $D$, AdamW optimizer,threshold
**Output:** Optimized weight matrices $W_0, W_1, \ldots, W_d$
Optimized bias vectors $b_1, \ldots, b_d$ Final output matrix $O$
**Initialize:**
 - Initialize weight matrices $W_0, W_1, \ldots, W_d$ with Gaussian distribution.
 - Initialize bias vectors $b_1, \ldots, b_d$ randomly.

**Training Loop:**
 **while** *Convergence criteria not met* **do**
   Initialize the output,$A_0 = W_0$
   **Forward Pass:**
   **for** *layer* $i = 1 \ldots d$ **do**
   | Compute $A_i = A_{i-1} \odot_{\max} W_i \oplus b_i$
   **end**
   **Loss Calculation:**
   Compute the loss $L = ||(A \odot_{\max} A^T)^{\text{off}} - D||_F^2$.
   **Backpropagation:**
   Compute gradients of $L$ with respect to $W_i$ and $b_i$ for each layer.
   **Parameter Update:**
   Use AdamW optimizer to update $W_i$ and $b_i$ based on the computed gradients.
   **Regularization:**
   **for** *layer* $i = 1 \ldots d$ **do**
   | Apply non-negativity constraints: $W_i \geq 0, b_i \geq 0$.
   **end**
   **Check for Convergence:**
   If $L$ change between iterations is smaller than the threshold, break the loop.
**end**

---

biases $b_1, \ldots, b_d$ are initialized using values drawn from a Gaussian distribution, where the mean is set to $\frac{\mu_D}{2d+2}$, with $\mu_D$ being the mean of the target matrix $D$ and $d$ representing the number of layers. The input matrix $W_0$ is initialized as a learnable non-negative matrix. During training, $X$ is propagated through the network, where tropical inner products and tropical addition are applied at each layer. The loss is calculated as the Frobenius norm of the difference between $(O \odot_{\max} O^T)^{\text{off}}$ and the target matrix $D$. Gradients of the loss are computed through backpropagation and the parameters are updated using AdamW. To maintain the matrix's metric properties, non-negativity constraints are enforced on each parameter after every update.

Through this approach, MetricEmbedding leverages tropical algebra to ensure that the learned matrix adheres to essential metric properties, while maintaining high computational efficiency. By incorporating a multi-layer architecture, MetricEmbedding captures complex structural patterns and balances expressiveness with robustness, overcoming the limitations of single-matrix representations.

### 3.3. Troptical innerproduct for Metric Learning

Traditional metric learning relies on Euclidean distance or cosine similarity(Bertinetto et al., 2016; Schroff et al., 2015; Kaya & Bilge, 2019), but these metrics struggle with capturing complex, non-linear relationships and may fail to satisfy the triangle inequality, which could negatively affect the performance of tasks such as contrastive learning(Khosla et al., 2020; Chen et al., 2020).To overcome these limitations, we introduce **MetricPlug**, a plugin that leverages tropical innerproduct to extend traditional metrics. Our approach ensures that the output satisfies metric properties and enables learning of more diverse metric relationships.

Once the embeddings $A$ are obtained from the neural network, we apply the ReLU activation function to ensure all elements are non-negative, which is a common requirement in metric learning. After applying the ReLU activation function to the embeddings $A$, the distance between any two points $i$ and $j$ in the learned space can be computed using the tropical inner product. The distance matrix is computed as:

$$O = \left( \text{ReLU}(A) \odot_{\max} \text{ReLU}(A)^T \right)^{\text{off}}$$

According to theorem 6, this ensures that the output matrix satisfies the metric properties, particularly non-negativity and the triangle inequality.

For a pair of embedding matrices $A$ and $B$ (both of size $N \times K$), the tropical distance between them is computed as:

$$O = \left( \text{ReLU}(A) \odot_{\max} \text{ReLU}(B)^T \right)^{\text{off}}$$

This is a submatrix of the larger tropical matrix $\left( \begin{bmatrix} ReLU(A) \\ ReLU(B) \end{bmatrix} \odot_{\max} \begin{bmatrix} ReLU(A) \\ ReLU(B) \end{bmatrix}^T \right)^{\text{off}}$.which is a valid metric matrix and preserves the metric properties required for distance computations.

The tropical distance matrix $O$ can be used in the same way as a traditional distance matrix for downstream tasks. For example, it can be used to compute the pairwise distances between data points, where $O_{ij}$ represents the tropical distance between points $i$ and $j$. This matrix can then be utilized in clustering, retrieval, or other metric-based tasks, similar to how Euclidean or cosine distance matrices are used. Additionally, $O$ preserves the metric properties, ensuring consistency in the relationship between data points.

### 3.4. Method Analysis

As summarized in Appendix C, MetricEmbedding significantly outperforms traditional methods in both time and space complexity. While methods such as TRF (Brickell et al., 2008) and PAF(Sonthalia & Gilbert, 2022) suffer from cubic time and space complexity ($O(n^3)$), MetricEmbedding reduces the time complexity to $O(n^2 k)$ and the

space complexity to $O(n^2)$, where $n$ denotes the number of points and $k$ is the embedding dimension. This improvement allows it to efficiently handle large-scale tasks, even when dealing with massive datasets that would overwhelm traditional matrix-based approaches.

One of the key advantages of MetricEmbedding lies in its use of tropical inner products to maintain the triangular inequality structure. This not only reduces the computational cost but also enables the algorithm to process larger matrices. More importantly, tropical inner product operations—based on max and addition—are naturally suitable for parallel computation on GPUs, which brings substantial performance gains. This parallelism makes MetricEmbedding particularly effective for large-scale data. By combining tropical algebra with mini-batch processing, MetricEmbedding efficiently performs large-scale matrix computations, making it well-suited for environments with limited memory resources.

Furthermore, the flexibility of MetricEmbedding allows for more adaptable weight updates. By calculating the loss function and applying gradient descent, it offers a dynamic approach to modifying results, unlike static methods. This flexibility is further enhanced by its support for online learning, enabling the algorithm to update continuously in real-time applications without needing to retrain from scratch. Such adaptability makes MetricEmbedding ideal for scenarios with evolving datasets, like recommendation systems or path optimization tasks.

## 4. Experiment

### 4.1. Experimental Setup

**Datasets:** The experiment uses synthetic data generated from a complete graph. The graph contains nodes of sizes 100, 500, 1000, with edge weights sampled from a uniform distribution $U(0, 1)$, following the data generation strategy outlined in the previous work (Li et al., 2023).

**Baselines:** We compare our method with three well-established algorithms:

- **TRF**: Efficient triangle fixing algorithm based on an iterative projection method, a classic algorithm (Brickell et al., 2008).
- **HLWB**: Efficient algorithm based on HLWB projections, a well-regarded method for metric nearness(Li et al., 2023).
- **PAF**: Project and Forget—an active set method for efficiently solving the metric nearness problem.(Sonthalia & Gilbert, 2022).

**Evaluation Metrics:** The following metrics are used to evaluate the performance of the methods:

- **Computation Time (s)**: The time required to compute the metric for the given matrix.
- **Prediction Accuracy**: We use the normalized mean squared error (NMSE) as the measure of nearness, defined as:

$$\text{NMSE} = \frac{\|X - D_o\|_F^2}{\|D_o\|_F^2}$$

where $X$ is the result obtained by applying different methods to process $D_o$, as in previous work (Li et al., 2023).

- **Triangle Inequality Violations (%)**: The percentage of triangle inequality violations after a fixed number of iterations. For convenience, the violation ratio is computed as $\frac{\text{count of violated inequalities}}{N^3}$

**Experimental Environment:** The experiments are conducted on a machine with an RTX 4090D GPU (24GB) and 32 vCPUs (Intel Xeon Platinum 8474C), running on a Linux-based operating system.

**Hyperparameters:** For MetricEmbedding, the learning rate is set to 0.001 using the AdamW optimizer. The model is initialized as described earlier. We use Python implementations of TRF and HLWB, and the Julia implementation of PAF.[1] All methods are evaluated based on their convergence time.

### 4.2. Performance evaluation

As shown in Table 1, our method (MetricEmbedding) outperforms TRF and HLWB in terms of computational efficiency, significantly reducing computation time even for large datasets. For instance, when $N = 100$, MetricEmbedding takes only 0.069 seconds, compared to 12.09 seconds for TRF and 14.80 seconds for HLWB. As the number of nodes increases, this time difference becomes even more pronounced, demonstrating our method's superior scalability. In terms of error rates, MetricEmbedding performs similarly to TRF,HLWB and PAF, maintaining a low error rate (e.g., 0.084 for $N = 100$), while offering a substantial reduction in computation time. Furthermore, MetricEmbedding consistently adheres to metric properties, with no triangle inequality violations, unlike TRF, which shows violations exceeding 4% on larger datasets. These results highlight the optimal balance between accuracy, computational efficiency, and metric preservation that MetricEmbedding achieves.

We tested the scalability of our approach, and although the computing platforms were not identical, the results highlight its superiority. Compared to TRF (Brickell et al., 2008) and PAF (Sonthalia & Gilbert, 2022), HLWB (Li et al., 2023) has a lower space complexity of $O(N^2)$, allowing it to handle larger matrices (e.g., over 10,000 points). However, its

---

[1]The recently open-sourced Python C implementation of HLWB may offer improved runtime performance.

*Table 1.* Comparison of Methods: Ours, TRF, HLWB, and PAF for Different Matrix Sizes. The table compares computation time, NMSE, and triangle inequality violations for four methods: Ours, TRF, HLWB, and PAF.

| Matrix Size (N) | Method | Computation Time (s) | NMSE (Ratio) | Triangle Inequality Violations (%) |
|---|---|---|---|---|
| 100 | Ours | 0.69 | 0.084 | 0% |
| | HLWB | 14.80 | 0.072 | 0% |
| | TRF | 12.09 | 0.059 | 4.71% |
| | PAF | 21.844 | 0.071 | 0% |
| 500 | Ours | 16.39 | 0.099 | 0% |
| | HLWB | 1291.61 | 0.069 | 0% |
| | TRF | 1120.73 | 0.058 | 4.53% |
| | PAF | 266 | 0.069 | 0% |
| 1000 | Ours | 26.73 | 0.136 | 0% |
| | HLWB | >2000 | 0.068 | 0% |
| | TRF | >2000 | 0.058 | 4.85% |
| | PAF | 1619.68 | 0.068 | 0% |

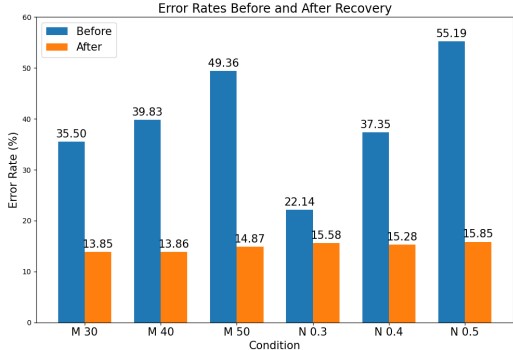

*Figure 2.* Performance comparison of the model under different missing rates and noise levels. "M30" indicates a missing rate of 30%, "N0.3" indicates Gaussian noise with a mean of 0.3, and similarly for other values.

time complexity limits its ability to scale. In contrast, our method achieves both optimal space complexity ($O(N^2)$) and time complexity ($O(N^2k)$), enabling it to efficiently handle much larger matrices. It can solve problems of size $\geq 100,000$ points within 12 hours, significantly outperforming HLWB, which can only handle matrices up to 10,000 points. This demonstrates the scalability and efficiency of our approach for large-scale problems.

### 4.3. Robustness Test

In this work, we evaluate the performance of the model in recovering incomplete matrices and matrices with noise, focusing on the impact of missing data and noise interference on recovery accuracy. In this experiment, we first computed the Euclidean distances between the points , generating the original distance matrix.To simulate incomplete data and noise interference, we introduced different missing rates

(30% to 50%) and noise levels (0.3 to 0.5). In the following figures, the notation "M30" represents a missing rate of 30%, "N0.3" represents Gaussian noise with a mean of 0.3, and similarly for other values. We applied missing data handling by randomly selecting data points and then used the MetricEmbedding model for recovery. Hyperparameter settings are the same as in Section 4.1. We evaluate the reconstruction quality using the Normalized Mean Squared Error (NMSE) defined in Section 4.2, which we refer to as the error rate.

As shown in Figure 2, the model performs reasonably well in both scenarios. Specifically, for both missing data and noisy data, the model significantly reduces the error and maintains low recovery error even in cases with high noise or high missing rates. These experimental results validate the effectiveness of the proposed algorithm in these complex scenarios.

### 4.4. Online Update Capability

To validate MetricEmbedding's online update capability, we incrementally fed new data and measured both update speed and prediction accuracy by varying the percentage of point pairs used for updates (from 20% to 100%).To test its online learning ability, new data was generated specifically for testing purposes.As shown in figure 3, The prediction error, defined as $\frac{||A-B||_F^2}{||B||_F^2}$, showed that the error stabilized as more data was added, improving from 88.32% to 67.33% as the update percentage increased. These results demonstrate MetricEmbedding's ability to maintain high accuracy and efficiency with relatively few updates, while handling complex relationships and preserving the underlying metric structure, even with noisy or multi-source data. Unlike traditional methods like PAF, which struggle with generalization, MetricEmbedding's multi-layer structure and tropi-

cal operations enable it to adaptively capture higher-order interactions, making it well-suited for dynamic tasks such as multi-source data alignment, recommendation systems, and metric learning.

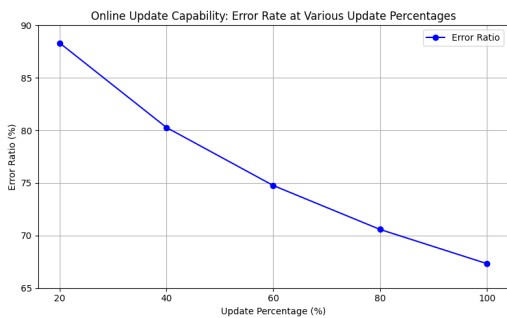

*Figure 3.* Online Update Capability: Error Rate at Various Update Percentages

### 4.5. Effect of Model Parameters and Latent Dimension (k) on Prediction Performance

We investigated the impact of model complexity, focusing on the latent dimension ($k$) and the number of layers ($D$) in the MetricEmbedding model. We evaluate the model performance using the normalized mean squared error (NMSE). Increasing $k$ improves model performance, as seen in the reduction of error from 95.28% to 88.33% when $k$ increases from 3 to 10. Moreover, increasing the model depth ($D$) enhances expressiveness, with error decreasing from 95.28% to 76.31% as the number of layers increases from 1 to 3. These results demonstrate a trade-off between model complexity and accuracy, where higher $k$ and deeper models lead to better performance but with increased computational costs. Despite this, MetricEmbedding maintains strong scalability, effectively handling high-dimensional data and complex relationships, making it suitable for real-time and large-scale applications.Detailed result in **Appendix D**.

### 4.6. MetricEmbedding for Distance Embedding

We assess MetricEmbedding's ability to generate distance embeddings, crucial for applications like clustering and recommendation systems.Unlike MDS(Carroll & Arabie, 1998) and PCA(Abdi & Williams, 2010), our method ensures that the recovered embeddings still satisfy the triangle inequality and maintain the properties of a metric. However, we do not directly compare with these methods. Instead, we conduct tests on datasets with different types of distance metrics, including Euclidean(Danielsson, 1980), Manhattan(Malkauthekar, 2013), Cosine similarity(Li & Han, 2013)and shortest-path (SP) distances(Potamias et al., 2009), to demonstrate the robustness and versatility of our approach. The results of the experiments, where the model

*Table 2.* MetricEmbedding Performance Across Different Distance Matrices

| Distance Matrix | Number of Points | Error (%) |
|---|---|---|
| Euclidean | 100 points | 2.49% |
| Euclidean | 1000 points | 2.61% |
| Manhattan | 100 points | 3.76% |
| Manhattan | 1000 points | 3.81% |
| Hamming | 100 points | 8.88% |
| Hamming | 1000 points | 9.05% |
| Cosine similarity | 100 points | 8.89% |
| Cosine similarity | 1000 points | 9.33% |
| Shortest-path | 100 points | 6.13% |
| Shortest-path | 1000 points | 2.42% |

was trained to predict distance matrices using embeddings of size $k = 3$, are summarized in the following table: These results demonstrate that MetricEmbedding efficiently generates accurate embeddings for large and diverse distance matrices, significantly reducing the computational cost of storing and querying distances. With space complexity of $O(Nk)$, where $N$ is the number of points and $k$ is the embedding dimension, MetricEmbedding outperforms traditional methods that require $O(N^2)$ storage. The model's ability to reconstruct the full distance matrix with only $O(N^2k)$ complexity makes it highly scalable for large datasets. These characteristics make MetricEmbedding an ideal solution for applications like clustering and recommendation systems, where fast and scalable distance computations are crucial.

### 4.7. Graph Contrastive Learning with MetricPlug

To evaluate the effectiveness of the MetricPlug method proposed in Section 3.3, we conducted an experiment based on graph contrastive learning—an unsupervised framework that learns node representations using contrastive loss. Specifically, we used the approach from GRACE (Zhu et al., 2020), which involves perturbing edges to generate augmented views. Unlike traditional methods that use cosine or Euclidean similarity metrics, we replaced them with MetricPlug, which uses the tropical inner product to calculate node pair similarity. The evaluation task focuses on node classification.

We used the **Cora** dataset, which contains 2708 nodes, and followed the standard data split used in GCN (Kipf & Welling, 2017). We compared our MetricPlug method with existing methods based on cosine, Hamming, Euclidean, and Manhattan distances.

We utilized the `dropout_adj` function in PyG to randomly perturb edges with a perturbation ratio of 0.1 to generate augmented views. The configuration used a learn-

ing rate of 0.01, 1000 epochs, with hidden and projection dimensions set to 64. **Accuracy** was the evaluation metric. The results are summarized in the table below:

*Table 3.* Comparison of different similarity metrics on the Cora dataset. MetricPlug achieves the best performance.

| Method | Validation Acc(%) | Test Acc(%) |
|---|---|---|
| Cosine | 77.4 | 76.5 |
| Manhattan | 79.0 | 79.2 |
| Euclidean | 78.2 | 79.0 |
| Hamming | 78.8 | 79.0 |
| **MetricPlug** | **79.2** | **79.4** |

As shown in the table 3, the MetricPlug outperforms other distance-based methods, achieving the best results on both validation and test sets.

## 5. Related work

The Metric Nearness Problem aims to find a distance matrix that satisfies metric constraints while minimizing distortion from a dissimilarity matrix(Sra et al., 2004). A naive approach uses QP, LP, or convex programming, but its high computational cost makes it impractical for large-scale problems. Exploiting triangle inequalities is key to improving efficiency.Sra et al. improved this by introducing the Triangle-Fixing (TF) algorithm, which has $O(n^3)$ complexity, but it still faced scalability issues (Sra et al., 2004). Brickell et al. extended this work with Bregman projections, enhancing convergence properties, yet their iterative process remained computationally expensive(Brickell et al., 2008).

Sonthalia and Gilbert's Project and Forget (PAF) algorithm(Sonthalia & Gilbert, 2022) offered linear convergence and a more dynamic approach to handling constraints, but it remains limited by the need for fixed input formats and struggles with real-time updates.Other methods, such as DeepNorm(Pitis et al., 2020), apply neural networks to map noisy matrices to valid distance matrices. While these approaches offer greater flexibility, they depend heavily on the quality of input data and still face challenges in terms of scalability. Additionally, isometrically embeddable matrices combined with the HLWB projection algorithm(Li et al., 2023) have improved scalability and convergence. However, these methods still face challenges in handling large-scale data and require significant computational resources, making them less effective in real-world complex scenarios.

Our approach, MetricEmbedding, leverages tropical algebra to ensure that metric properties are preserved while improving computational efficiency. By eliminating the need for complex projections and enabling real-time updates, MetricEmbedding provides a more scalable and flexible solution

for large-scale tasks, overcoming many of the limitations of previous methods.

Beyond the foundational approaches, other aspects of the metric repair problem have also received considerable attention. Gilbert and Jain (Gilbert & Jain, 2017) introduced the sparse metric repair problem, proposing several algorithms to enforce metric consistency with minimal modifications. Fan et al. (Fan et al., 2020) extended this framework to graph structures and proposed a model for generalized metric repair that is suitable for settings with partially observed or incomplete data. Fan, Raichel, and Van Buskirk (Fan et al., 2018) proved that the increase-only and general variants of the Metric Violation Distance (MVD) problem are NP-complete, and developed approximation algorithms for these cases. Additionally, from a fitting perspective, Cohen-Addad et al. (Cohen-Addad et al., 2022) studied how to approximate arbitrary distance data using tree-metric structures, and proposed algorithms with constant-factor guarantees, applicable in clustering and phylogenetic analysis. In addition, Laub et al. (Schölkopf et al., 2007) showed through psychophysical experiments that human similarity judgments often violate metric properties, suggesting that non-metricity in real-world data may stem from perceptual or cognitive limitations.

## 6. Conclusion

In this paper, we address the Metric Nearness Problem by introducing a novel theoretical framework that connects tropical inner products with metric matrices. The TA-metric representation theorem reveals that tropical inner products can represent matrices satisfying the triangle inequality. Based on this, we propose MetricEmbedding, a network architecture that efficiently predicts metric matrices, offering improvements in computational efficiency, scalability, and real-time prediction. Experimental results demonstrate that our method achieves up to 60× speedup over state-of-the-art approaches, while handling larger matrices with lower memory usage. In addition, we propose MetricPlug, a tropical inner product–based method for efficiently computing distance matrices that satisfy metric properties while capturing complex relationships beyond traditional metrics. Its effectiveness is validated through experiments within a graph contrastive learning framework.

## Acknowledgement

This work was supported in part by the National Key Research and Development Program of China (No. 2024YDLN0005). W. Wang was supported by Guangdong Provincial Key Lab of Integrated Communication, Sensing and Computation for Ubiquitous Internet of Things (No.2023B1212010007, SL2023A03J00934),

Guangzhou Municipal Science and Technology Project (No. 2023A03J0003, 2023A03J0013 and 2024A03J0621). In addition, we thank Dr Linshan Jiang's comment and help on this work.

## Impact Statement

Our research introduces a novel theoretical and computational framework based on the tropical inner product, opening new avenues for metric learning. The proposed MetricEmbedding architecture demonstrates strong performance in scalability, efficiency, and real-time prediction, with broad application prospects. Meanwhile, the Metric-Plug method shows promise in advancing contrastive and metric learning.Besides, we do not think ourwork will have a bad impact on ethical aspects and futuresocietal consequences.

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

# A. Hierarchical Graph and Tropical Inner Product

## A.1. three levels graph and Tropical Inner Product

Consider a graph divided into three levels:

- $V_1$ is the first set of nodes in the graph,

- $V_2$ is the second set of nodes in the graph,

- $V_3$ is the third set of nodes in the graph.

We have two matrices:

- **Matrix 1**: Represents the edge weights from $V_1$ to $V_2$, where the element $E_1^{ij}$ denotes the edge weight from node $i \in V_1$ to node $j \in V_2$,

- **Matrix 2**: Represents the edge weights from $V_2$ to $V_3$, where the element $E_2^{ij}$ denotes the edge weight from node $i \in V_2$ to node $j \in V_3$.

If we perform the tropical inner product of Matrix 1 and Matrix 2, i.e.,

$$C_1 = E_1 \odot_{\max} E_2, \quad C_2 = E_1 \odot_{\min} E_2$$

then $C_1$ and $C_2$ are new matrices that represent the paths from nodes in $V_1$ to nodes in $V_3$. Specifically, each element $C_{ij}$ in matrix $C$ represents the path weight from node $i \in V_1$ to node $j \in V_3$, where $C_1$ corresponds to the maximum path weight (using the $\max$ operation) and $C_2$ corresponds to the shortest path weight (using the $\min$ operation).

## A.2. Multi-layer Graph Extension

Consider a graph consisting of multiple layers, where each layer's node set is represented as $V_1, V_2, V_3, \ldots, V_k$. Each matrix represents the edge weights from one layer to the next:

- **Matrix 1**: Represents the edge weights from $V_1$ to $V_2$,

- **Matrix 2**: Represents the edge weights from $V_2$ to $V_3$,

- $\ldots$

- **Matrix (k-1)**: Represents the edge weights from $V_{k-1}$ to $V_k$.

Assume we have $k$ layers, corresponding to matrices $E_1, E_2, \ldots, E_{k-1}$. By performing consecutive tropical inner products, we can compute the path information from $V_1$ to $V_k$.

First, we perform the tropical inner product between $E_1$ and $E_2$ to obtain a new matrix $C_1$:

$$C_1 = E_1 \odot_{min} E_2$$

Each element $C_{ij}$ in matrix $C_1$ represents the minimum path weight from node $i \in V_1$ to node $j \in V_3$.

Next, we perform the tropical inner product between $C_1$ and $E_3$, obtaining matrix $C_2$:

$$C_2 = C_1 \odot E_3$$

Each element $C_{ij}$ in matrix $C_2$ represents the minimum path weight from node $i \in V_1$ to node $j \in V_4$.

By continuing this process, we multiply matrix $C_2$ by $E_4$ to obtain matrix $C_3$:

$$C_3 = C_2 \odot E_4$$

This process is repeated until all layers have been considered. The final matrix $C_{k-1}$ will contain the path information from $V_1$ to $V_k$.

Each element $C_{ij}$ in matrix $C_i$ represents the minimum path weight from node $i \in V_1$ to node $j \in V_k$. The same applies for the maximum path weight when the tropical inner product is defined based on the $\max$-plus operation.

## B. Additional Proofs and Theoretical Framework

**Theorem 1:** If matrix $A \in \mathbb{R}^{N \times N}$ satisfies the triangle inequality, and matrix $B \in \mathbb{R}^{N \times N}$ satisfies the triangle inequality, then the matrix $C = A + B \in \mathbb{R}^{N \times N}$ also satisfies the triangle inequality.

**Proof:** For any indices $i, j, k$, since $A$ and $B$ satisfy the triangle inequality, we have:

$$A_{ij} + A_{jk} \geq A_{ik}, \quad B_{ij} + B_{jk} \geq B_{ik}.$$

Adding these two inequalities:

$$A_{ij} + A_{jk} + B_{ij} + B_{jk} \geq A_{ik} + B_{ik}.$$

This implies:

$$C_{ij} + C_{jk} \geq C_{ik}.$$

Thus, $C = A + B$ satisfies the triangle inequality.

**Theorem 2:** If $A$ is a metric matrix, then for any positive constant $\alpha$, $\alpha A$ is also a metric matrix.

**Proof:** Since $A$ satisfies the triangle inequality $A_{ik} + A_{kj} \geq A_{ij}$, multiplying both sides by $\alpha > 0$ preserves the inequality:

$$\alpha A_{ik} + \alpha A_{kj} \geq \alpha A_{ij}.$$

Thus, $\alpha A$ satisfies the triangle inequality, is symmetric, and has diagonal elements equal to zero. Hence, $\alpha A$ is a metric matrix.

**Theorem 3:** If a matrix $A$ satisfies the triangle inequality, then the matrix $A + A^T = B$, and after setting the diagonal elements of $B$ to 0, the resulting matrix satisfies the metric properties.

**Proof:** By Theorem 1, $A + A^T$ satisfies the triangle inequality and is symmetric. Setting its diagonal elements to 0 preserves the triangle inequality, and the matrix remains symmetric. Thus, the modified matrix $B$ satisfies the conditions for a metric matrix.

thus,we just need to care the relationship between tropical algebra and triangle inequality.

**Theorem 4:** If $A$ and $B$ are both $N \times N$ metric matrices, then the element-wise maximum of $A$ and $B$, denoted as $\max(A, B)$, is also a metric matrix.

**Proof**: Suppose $A$ and $B$ are metric matrices, meaning that for all $i, j, k$, we have:

$$A_{ik} + A_{kj} \geq A_{ij}, \quad B_{ik} + B_{kj} \geq B_{ij}.$$

Let $C = \max(A, B)$. For any $i, j, k$, we have:

$$C_{ik} + C_{kj} = \max(A_{ik}, B_{ik}) + \max(A_{kj}, B_{kj}).$$

By the properties of the maximum function, this implies:

$$C_{ik} + C_{kj} \geq \max(A_{ik} + A_{kj}, B_{ik} + B_{kj}) \geq \max(A_{ij}, B_{ij}).$$

Hence, $C_{ik} + C_{kj} \geq C_{ij}$, proving that $C$ is a metric matrix.

**generalization of Theorem 5:** Let $A \in \mathbb{R}^{N \times K}$ and $B \in \mathbb{R}^{K \times M}$ be non-negative matrices, and let $C = A \odot_{\max} B$. For any $i \in \{1, \ldots, N\}, k \in \{1, \ldots, min(N, M)\}$, and $j \in \{1, \ldots, M\}$, we have:

$$C_{ik} + C_{kj} \geq C_{ij}.$$

This inequality defines the **tropical triangle inequality** for matrices. Specifically, when $N = M$, this becomes the tropical triangle inequality for square matrices.

**Proof**: By definition, $C_{ij} = \max_r(A_{ir} + B_{rj})$. Assume that when $r = s$, equality holds, i.e., $C_{ij} = A_{is} + B_{sj}$. Then:

$$A_{is} \leq A_{is} + B_{sk} \leq C_{ik} = \max_r(A_{ir} + B_{rk}),$$

and

$$B_{sj} \leq A_{ks} + B_{sj} \leq C_{kj} = \max_r(A_{kr} + B_{rj}).$$

Thus, we have:

$$C_{ij} \leq C_{ik} + C_{kj}.$$

Therefore, the tropical inner product satisfies the tropical triangle inequality.

**Theorem 6:** Let $A$ be a non-negative matrix of size $N \times K$. Define $B = A \odot_{\max} A^T$ as the tropical inner product of $A$ with its transpose. Let $B^{\text{off}}$ be the matrix obtained by setting the diagonal elements of $B$ to zero. Then, $B^{\text{off}}$ is a metric matrix.

**Proof**: From Theorem 5, we know that $A \odot_{\max} A^T$ satisfies the triangle inequality. Now, we will prove that $B$ is symmetric and satisfies the properties of a metric matrix.

1. Symmetry: For any $i, j$, we have:

$$B_{ij} = \max_k(A_{ik} + A_{jk}^T) = \max_k(A_{ik} + A_{kj}),$$

$$B_{ji} = \max_k(A_{jk} + A_{ki}^T) = \max_k(A_{jk} + A_{ik}),$$

so $B_{ij} = B_{ji}$. Therefore, $B$ is symmetric.

2. diagonal elements: Since $B^{\text{off}}$ is obtained by setting the diagonal elements of $B$ to zero, it follows that:

$$B_{ii}^{\text{off}} = 0 \quad \text{for all } i.$$

Therefore, $B^{\text{off}}$ is a symmetric matrix with zero diagonal elements and satisfies the properties of a metric matrix.

**Theorem 7:** Let $A \in \mathbb{R}^{N \times k}$ and $B \in \mathbb{R}^{k \times M}$ be non-negative matrices, where $k > N * M$. Then, there exist matrices $A' \in \mathbb{R}^{N \times k'}$ and $B' \in \mathbb{R}^{k' \times N}$, with $k' \leq N * M$, such that:

$$A' \odot_{\max} B' = A \odot_{\max} B.$$

**Proof:** Consider the matrix $C = A \odot_{\max} B$, where $C \in \mathbb{R}^{N \times M}$. The $(i, j)$-th element of $C$ is given by:

$$C_{ij} = \max_k(A_{ik} + B_{kj}).$$

Let $\{k\}$ denote the index set for which the equality $C_{ij} = A_{ik} + B_{kj}$ holds. Since the maximum is taken over $k$, the set $\{k\}$ can contain at most $N * M$ distinct indices, as there are at most $N * M$ possible unique combinations of rows of $A$ and columns of $B$.

Thus, we can construct new matrices $A'$ and $B'$ by retaining only the corresponding $k$-th columns of $A$ and the $k$-th rows of $B$, respectively. Specifically, $A' \in \mathbb{R}^{N \times k'}$ and $B' \in \mathbb{R}^{k' \times M}$ are defined by selecting the relevant columns and rows corresponding to the set $\{k\}$, where $k' \leq N * M$. By this construction, it follows that:

$$A' \odot_{\max} B' = A \odot_{\max} B,$$

proving the result.

**Theorem 8: max-TA-metric Representation Theorem**

Let $\Omega$ denote the set of all matrices produced by tropical inner products of non-negative matrices:

$$\Omega = \left\{ \mathbf{M} \, \middle| \, \mathbf{M} = (\mathbf{X} \odot_{max} \mathbf{Y}^T)^{\text{off}}, \ \mathbf{X}, \mathbf{Y} \in \mathbb{R}_{\geq 0}^{N \times K} \right\},$$

where $(\mathbf{X} \odot_{max} \mathbf{Y}^\mathbf{T})_{ij} = \max_k(\mathbf{X}_{ik} + \mathbf{Y}_{jk})$.

Let $\Psi \subset \mathbb{R}_{\geq 0}^{N \times N}$ be the set of matrices $\mathbf{D}$ satisfying:

- $\mathbf{D}_{ii} = 0 \quad \forall i \in \{1, 2, ..., N\}$,

- $\mathbf{D}_{ij} \leq \min_{k_1 \neq i} \mathbf{D}_{ik_1} + \min_{k_2 \neq j} \mathbf{D}_{k_2 j} \quad \forall i, j \in \{1, 2, ..., N\}$,

Then, $\Omega = \Psi$. The set of matrices generated via the tropical inner product (with diagonals removed) from non-negative matrices constitutes a specific subset of the set of non-negative matrices with zero diagonals that satisfy the triangle inequality.

**Proof:**

1. For any $M \in \Omega$, there exist non-negative matrices $X, Y \in \mathbb{R}_{\geq 0}^{N \times K}$ such that:

$$M_{ij} = \max_k (X_{ik} + Y_{jk}) \quad \text{for } i \neq j, \quad M_{ii} = 0.$$

For the case where $i = j$, we have $M_{ii} = 0$, and

$$M_{ij} \leq \min_{k_1 \neq i} \mathbf{M}_{ik_1} + \min_{k_2 \neq j} \mathbf{M}_{k_2 j}.$$

For the other cases, choose some $k$ such that:

$$M_{ij} = X_{ik} + Y_{jk}.$$

For any $p, q \in \{1, \ldots, n\}$ with $p \neq i$ and $q \neq j$, we have:

$$X_{ik} + Y_{qk} \leq M_{iq}, \quad X_{pk} + Y_{jk} \leq M_{pj}.$$

Adding the two inequalities gives:

$$M_{iq} + M_{pj} \geq X_{ik} + Y_{jk} = M_{ij}.$$

Thus, for any $p, q \in \{1, \ldots, n\}$ with $p \neq i$ and $q \neq j$, it follows that

$$\min_{p \neq i} M_{ip} + \min_{q \neq j} M_{qj} \geq M_{ij}.$$

Therefore, we conclude that $M \in \Psi$.

2. For any matrix $M \in \Psi$, we provide a constructive proof in the appendix, showing that there exist matrices $X, Y \in \mathbb{R}_{\geq 0}^{N \times N^2}$ such that:

$$M = (X \odot_{\max} Y)^{\text{off}}.$$

For any $i$ and $j$, we can construct two matrices $X$ and $Y$ such that:

$$t(i, j) = N \cdot (i - 1) + j, \quad X_{it(i,j)} + Y_{jt(i,j)} = M_{ij}.$$

To ensure $M_{ij} = X_{it(i,j)} + Y_{jt(i,j)}$, we further require that for any $k \neq i$, we have $X_{k,t(i,j)} = 0$, and for any $k \neq j$, we have $Y_{k,t(i,j)} = 0$. Additionally, we need to ensure the following inequalities:

$$\forall k \neq j, \quad X_{it(i,j)} \leq M_{ik}, \quad \forall k \neq i, \quad Y_{j,t(i,j)} \leq M_{kj}.$$

Since $M$ satisfies the inequality

$$\min_{k_1 \neq i} M_{ik_1} + \min_{k_2 \neq j} M_{k_2 j} \geq M_{ij},$$

we can conclude that such matrices $Y$ and $Z$ exist.

Specifically, we set:

$$X_{i,t(i,j)} = min(\min_{k \neq j} M_{i,k}, M_{i,j}) \geq 0, \quad Y_{j,t(i,j)} = M_{i,j} - X_{i,t(i,j)} \geq 0.$$

Thus, we can find matrices $X$ and $Y$ such that $M = (X \odot_{\max} Y^T)^{\text{off}}$, and we conclude that:

$$M \in \Omega$$

**Conclusion:** Thus, $\Omega = \Phi$.

**Theorem 9:min-TA-metric Representation Theorem** Let $\Omega$ be the set of all results of the min-tropical inner product (using max-plus operation) of two non-negative matrices (excluding the diagonal elements), and let $\Phi$ be the set of all non-negative matrices that satisfy the triangle inequality (excluding the diagonal elements). Then, $\Phi \subseteq \Omega$.

**Firstly, the min-Tropical Inner Product of Two Non-negative Matrices Does Not Satisfy the Triangle Inequality:**

Consider the non-negative matrices $A$ and $B$:

$$A = \begin{bmatrix} 0.1 & 0.9 & 0.5 \\ 0 & 0 & 0 \\ 1 & 1 & 1 \end{bmatrix}, \quad B = \begin{bmatrix} 0 & 0 & 0.9 \\ 0 & 0 & 0.1 \\ 0 & 0 & 0.5 \end{bmatrix}.$$

When $A_{sk} = B_{ks} = 0$, we show that the tropical inner product may not satisfy the triangle inequality. Consider the following terms:

For some values of $A$ and $B$, we have:

$$C_{12} + C_{23} < C_{13}.$$

Thus, the tropical inner product $\odot_{\min}$ does not always satisfy the triangle inequality in this case. In general, the relation

$$\min(A_{ir} + B_{rk}) + \min(A_{kr} + B_{rj}) \not\geq \min(A_{ir} + B_{rj})$$

shows that the triangle inequality does not hold universally for the min-tropical inner product.

**Next, Consider a Metric Matrix $C \in \Phi$:**

Now, for a matrix $C$ belonging to the set $\Phi$, which satisfies the triangle inequality, we have the following important results:

$$A \odot_{\min} A = A$$

**Proof:**

For any $i, j, k$, by the metric properties of $A$, we know that:

$$A_{ik} + A_{kj} \geq A_{ij}.$$

Thus, for any $k$, the following holds:

$$\min_k(A_{ik} + A_{kj}) \geq A_{ij}.$$

Furthermore, when $k = i$, utilizing the property $A_{ii} = 0$, we obtain:

$$\min_k(A_{ik} + A_{kj}) = A_{ij}.$$

Therefore, we conclude:

$$A \odot_{\min} A = A.$$

Additionally, if $A$ is a symmetric matrix ($A^T = A$), we also have:

$$A \odot_{\min} A^T = A.$$

**Conclusion:** This result shows that the min-tropical inner product of a matrix with itself yields the original matrix, and the symmetry of the matrix further confirms the consistency of the operation under the min-tropical product. This property holds for all metric matrices in $B$, meaning that the min-tropical inner product preserves the structure of metric matrices that satisfy the triangle inequality.

**Theorem 10:** The tropical inner product of a metric matrix with itself satisfies the triangle inequality.

**Proof:** When using the $\min$-sum operation, as shown in Theorem 9, the tropical inner product of a matrix with itself yields the matrix itself, which naturally satisfies the triangle inequality.

When using the $\max$-sum operation, the tropical inner product of two non-negative matrices satisfies the triangle inequality. However, the diagonal elements of the resulting matrix may not necessarily be zero, but the matrix still satisfies the triangle inequality.

**Theorem 11** Let $A$ be a non-negative matrix of dimension $m \times n$ and $B$ be a non-negative matrix of dimension $n \times p$. Define the following operation for any $i, j$:

$$d_{ij} = \min_s A_{is} + \min_s B_{sj},$$

where the minimum is taken over index $s$, i.e., for each pair $i, j$, we compute the minimum value of $A_{is}$ for all $s$ (with $s \in \{1, 2, \ldots, n\}$), and similarly for $B_{sj}$ (with $s \in \{1, 2, \ldots, n\}$).

Then, for any $i, j, k$, we have the following inequality:

$$d_{ik} + d_{kj} \geq d_{ij}.$$

**Proof:** By the definition of $d_{ij}$, we have:
$$d_{ij} = \min(A_{is}) + \min(B_{sj}).$$

Now, consider $d_{ik} + d_{kj}$:

$$d_{ik} + d_{kj} = \min(A_{is}) + \min(B_{sk}) + \min(A_{ks}) + \min(B_{sj}).$$

By the non-negativity of the matrices $A$ and $B$, we have:

$$d_{ik} + d_{kj} \geq \min(A_{is}) + \min(B_{sj}) = d_{ij}.$$

Thus, the triangle inequality holds, proving the result.

**Theorem 12:**
Let $A$ be a non-negative matrix of dimension $m \times n$, $B$ be a non-negative matrix of dimension $n \times p$, and $O$ be a zero matrix of appropriate dimension. Then, the operation $A \odot_{\min} O \odot_{\min} B$ satisfies the triangle inequality.

**Proof:** By definition, we have:

$$C_{ij} = \min_{s1,s2}(A_{i,s1} + 0 + B_{s2,k}) = \min_s(A_{is}) + \min_s(B_{sk}).$$

This relation satisfies the conditions for the triangle inequality. We can further prove it using a layered graph method.

**Theorem 13:**

Let $G(E, V)$ be an undirected graph with the following properties:

- There are no negative-weight cycles,

- There are no self-loops,

- All edge weights are non-negative,

- There is at most one edge between any two nodes,

- The graph is connected, i.e., there exists at least one path between any pair of nodes.

Let $A$ be the adjacency matrix of $G$, defined as follows:

$$A[i,j] = \begin{cases} 0 & \text{if } i = j, \\ G[i,j] & \text{if there is an edge between } i \text{ and } j, \\ \infty & \text{otherwise.} \end{cases}$$

Now, define the recurrence relation for $A^k$:

$$A^k = A^{k-1} \odot_{\min} A$$

where $\odot_{\min}$ denotes the min-tropical matrix multiplication(min-sum). Then, under tropical matrix multiplication, the matrix $A^{N-1}$ (the matrix obtained after applying tropical matrix multiplication $N-2$ times) satisfies the triangle inequality. In particular, $A^{N-1}$ represents the shortest path distance matrix for the graph $G$.

**Proof**

As shown in A.2 before, The tropical minimum operation (min-sum) computes the shortest path and weight sum between $N$ points on the left and $N$ points on the right. The matrix $A^K_{i,j}$ represents the shortest path weight sum between points $i$ and $j$ through at most $k$ edges. Since the graph $G$ has $N$ nodes, the shortest path between any two points will not exceed $N-1$ edges. Therefore, $A^{N-1}[i, j]$ gives the shortest path distance matrix for the graph.

For an undirected graph, the shortest path distances naturally satisfy the triangle inequality, making the matrix a metric matrix. For a directed graph, this can be ensured by taking $A^{N-1} + (A^{N-1})^T$, which satisfies the metric properties.

where $\odot_{\min}$ denotes the min-tropical matrix multiplication (min-sum). Then, under tropical matrix multiplication, the matrix $A^{N-1}$ (the matrix obtained after applying tropical matrix multiplication $N-2$ times) satisfies the triangle inequality. In particular, $A^{N-1}$ represents the shortest path distance matrix for the graph $G$.

By combining this theorem, we can further accelerate the algorithm for the complete graph. By combining tropical matrix decomposition techniques, we are able to achieve an $O(KN^2)$ approximation algorithm. That is, for a complete matrix $A$ of dimension $N \times N$, we approximate it as $B \odot B^T$, where $B$ is of dimension $N \times K$, and $K$ is a constant much smaller than $N$. Specifically, it can be written as

$$(B \odot_{\min} B^T)^{N-1} = B \odot (B^T \odot_{\min} B)^{N-2} B^T.$$

By using the fast exponentiation algorithm, the computation can be performed in $O(K^2 N \log N)$, while the $N \times K \odot K \times K$ matrix has a complexity of $O(Nk^2)$, and the $N \times K \odot K \times N$ has a complexity of $O(N^2k)$. Therefore, the overall computational complexity is $O(kN^2)$.

**Theorem 14:** Any non-negative real matrix, under the tropical inner product (either the $\min$ or $\max$ operation), will satisfy the triangle inequality after a finite number of powers.

**Proof:** In the case of the $\odot_{min}$ operation, by Theorem 11, it is known that the matrix will satisfy the triangle inequality after at most $N-1$ powers. This is because the shortest path between any two nodes in a graph with $N$ nodes cannot exceed $N-1$ edges, as discussed in the proof of Theorem 1. Hence, after at most $N-1$ powers, the matrix will represent the shortest path distances and satisfy the triangle inequality.

In the case of the $\odot_{max}$ operation, as described in Theorem 5, the tropical inner product of two non-negative matrices automatically satisfies the triangle inequality. Therefore, under the $\odot_{max}$ operation, the tropical inner product of any two matrices satisfies the triangle inequality.

Thus, regardless of whether the operation is $\odot_{min}$ or $\odot_{max}$, a finite number of powers (at most $N-1$) will yield a matrix that satisfies the triangle inequality. By making a slight adjustment, this matrix can further be made to satisfy the properties of a metric matrix.

# C. Supplementary Details on Metric Embedding

*Table 4.* Comparison of Computational Complexity, Scalability, and Execution Platform

| Method | Time Complexity | Space Complexity | Scalability |
|---|---|---|---|
| DeepNorm | — | — | Suitable for small-scale problems, runs on GPU |
| TRF Algorithm | $O(n^3)$ | $O(n^3)$ | Suitable for small-scale problems, runs on CPU |
| PAF Algorithm | $O(n^3)$ | $O(n^3)$ | Limited to small datasets, runs on CPU |
| HLWB Algorithm | $O(n^3)$ | $O(n^2)$ | Suitable for medium-scale data, runs on CPU |
| Tropical (Normal) | $O(n^2k)$ | $O(n^2)$ | Highly scalable, supports real-time updates, run on GPU |

## C.1. Method Analysis

Leveraging tropical algebra, MetricEmbedding ensures the learned matrix adheres to fundamental metric properties, such as the triangle inequality. Unlike methods like HLWB projections (Li et al., 2023), which rely on linear constraints or complex projections, MetricEmbedding's tropical operations inherently preserve these properties. This guarantees the correctness and robustness of the solution. In contrast, methods like TRF(Brickell et al., 2008) may violate metric properties during optimization, as they do not maintain metric consistency throughout the process.

As shown in Table 4, The tropical algebra operations in MetricNN provide significant computational efficiency, even for large matrices, addressing the limitations of traditional approaches like the TRF (Brickell et al., 2008), which exhibits $O(n^3)$ space complexity. MetricEmbedding reduces memory cost to $O(n^2)$, making it highly scalable and efficient for large-scale datasets. Additionally, its support for minibatch processing and dynamic point-pair weight updates ensures that it can handle large matrices with high efficiency—features that many existing methods, such as PAF (Sonthalia & Gilbert, 2022), cannot accommodate due to their reliance on fixed input formats and limited parallelization.

## C.2. Minibatch-Based Metric Embedding

Here, we present a minibatch-based implementation of MetricEmbedding. As shown in algorithm2,compared to the standard MetricEmbedding, the main modification is the integration of minibatch processing into the model, allowing it to overcome the limitation of storing $O(N^2)$ pairwise values. This enables the method to handle larger-scale data, with the minimum space complexity reduced to $O(Nk)$.

---

**Algorithm 2** Minibatch Training Procedure for MetricEmbedding

---

**Input:** Target matrix $D$, batch size $B$, AdamW optimizer, threshold
**Output:** Optimized weight matrices $W_0, W_1, \ldots, W_d$
Optimized bias vectors $b_1, \ldots, b_d$
Final output matrix approximation from minibatches
**Initialize:**
 - Initialize weight matrices $W_0, W_1, \ldots, W_d$ with Gaussian distribution.
 - Initialize bias vectors $b_1, \ldots, b_d$ randomly.
 - Split $D$ into minibatches, each containing $B$ randomly sampled point pairs $(i, j)$.

**Training Loop:**
 **while** *Convergence criteria not met* **do**
  **for** *each minibatch $\mathcal{B}$ containing pairs $(i, j)$* **do**
   **Forward Pass:**
    - Extract relevant rows $W_0^{(i)}, W_0^{(j)}$ for batch processing.
    - Initialize the output for batch: $A_0^{(i)}, A_0^{(j)} = W_0^{(i)}, W_0^{(j)}$.
    - **for** *layer $l = 1 \ldots d$* **do**
      Compute batch-wise embedding:
      $$A_l^{(i)} = A_{l-1}^{(i)} \odot_{\max} W_l \oplus b_l$$
      $$A_l^{(j)} = A_{l-1}^{(j)} \odot_{\max} W_l \oplus b_l$$

    **end**
   **Loss Calculation:**
    Compute loss only for batch elements:

    $$L = \sum_{(i,j) \in \mathcal{B}} \left\| \left( A_d^{(i)} \odot_{\max} (A_d^{(j)})^T \right) - D_{ij} \right\|_F^2$$

   **Backpropagation:**
    Compute gradients of $L$ with respect to $W_l$ and $b_l$ using only batch data.
   **Parameter Update:**
    Use AdamW optimizer to update $W_l$ and $b_l$ based on computed gradients.
   **Regularization:**
    **for** *layer $l = 1 \ldots d$* **do**
     Apply non-negativity constraints: $W_l \geq 0$, $b_l \geq 0$.
    **end**
  **end**
  **Check for Convergence:**
   If $L$ change between iterations is smaller than the threshold, break the loop.
 **end**

---

*Table 5.* Comparison of Space Complexity, Time Complexity, and Max Problem Size.*Note: Data for TRF, PAF, and HLWB were taken from the respective studies: (Li et al., 2023).*

| Algorithm | Largest Size (n) |
|---|---|
| CPLEX | $< 300$ |
| MOSEK | $< 300$ |
| TRF(Brickell et al., 2008) | $< 2,000$ |
| PAF(Sonthalia & Gilbert, 2022) | $\sim 3,000$ |
| DeepNorm(Pitis et al., 2020) | $< 1,500$ |
| HLWB(Li et al., 2023) | $> 10,000$ |
| Tropical | $> 100,000$ |

# D. Experiment

## D.1. Comparison of Scalability with Existing Methods

We tested the scalability of our approach on a variety of problem sizes, and while the computing platforms were not identical, the results consistently highlight the superiority of our method. Specifically, compared to the TRF method (Brickell et al., 2008) and the PAF algorithm (Sonthalia & Gilbert, 2022), HLWB (Li et al., 2023) offers a lower space complexity of $O(N^2)$, enabling it to handle matrices with over 10,000 points. However, HLWB's time complexity remains a bottleneck, restricting its scalability to larger datasets.

In contrast, our method achieves both optimal space complexity ($O(N^2)$) and time complexity ($O(N^2 k)$), allowing it to efficiently scale to much larger matrices. Our approach can solve problems with sizes as large as 100,000 points within 12 hours, demonstrating a significant performance improvement over HLWB, which can only handle matrices up to 10,000 points. By adopting a minibatch-based approach, our method can even handle data at the scale of $10^6 \times 10^6$. Although the optimization performance may be limited by the problem size, our method can at least provide a solution that strictly satisfies the triangle inequality.This improvement underscores the scalability and efficiency of our approach, even for very large-scale problems that are common in real-world applications.

Table 5 summarizes the comparison between our method and existing approaches. The empirical results clearly demonstrate the advantages of our approach, making it an ideal solution for large-scale metric optimization problems.

## D.2. Online Update Capability

To validate the online update capability of MetricEmbedding, we incrementally fed new data into the model and evaluated the update speed and prediction accuracy by varying the percentage of point pairs used for updates (from 20% to 100%). To better emphasize the model's online learning ability, we specifically generated a new dataset with the following data generation strategy:

- **Data Distribution:** A distance matrix with significant variation was generated by intentionally clustering most of the points in one region, while a few points were placed far from this region. This distribution ensures that there is a larger variation in distances between points, which increases the challenge of the online learning task.

- **Data Generation Process:**
    - The majority of points (50 points) are clustered around the coordinates (50, 50), with the distribution following a standard normal distribution and a standard deviation of 5 to ensure they are concentrated near the center.
    - A few points (10 points) are distributed away from the clustered region (around coordinates (200, 200)), with their distribution also following a standard normal distribution and a standard deviation of 20, enhancing the distance differences between these points and the center.

- **Distance Matrix Calculation:** The Euclidean distances between these points were computed, and a distance matrix was generated for use in subsequent experiments.

*Table 6.* NMSE and Triangle Inequality Violation Ratio at different update percentages

| Update (%) | NMSE | TIV Ratio |
|---|---|---|
| 20 | 0.8832 | 0 |
| 30 | 0.8388 | 0 |
| 40 | 0.8028 | 0 |
| 50 | 0.7728 | 0 |
| 60 | 0.7476 | 0 |
| 70 | 0.7257 | 0 |
| 80 | 0.7058 | 0 |
| 90 | 0.6935 | 0 |
| 100 | 0.6773 | 0 |

By using this data generation strategy, we ensured that the distance differences between point pairs in the experiment were significant, which posed a greater challenge for the online learning task and tested the model's ability to adapt to varying levels of input data. The following table presents the normalized mean square error (NMSE) and the triangle inequality violation ratio(TIV Ratio) at different update percentages:

The table above presents the experimental results of the MetricEmbedding model at various update percentages, focusing on two key performance metrics: Normalized Mean Square Error (NMSE) and Triangle Inequality Violation Ratio (TIV Ratio).

**Normalized Mean Square Error (NMSE)**: The NMSE values progressively decrease as the update percentage increases. At the 20% update level, the NMSE is 0.8832, and it improves as more data is added. By the 100% update level, the NMSE reaches 0.6773, indicating that the model's performance improves as it adapts to more data. This trend shows that the model's ability to make accurate predictions increases with the number of updates, demonstrating its capacity to learn and adapt to new data in an online learning setting.

**Triangle Inequality Violation Ratio (TIV Ratio)**: Notably, the TIV Ratio remains at 0 for all update percentages. This indicates that the model consistently satisfies the triangle inequality throughout the updates, which is a critical property for distance-based models. The absence of violations suggests that the model is stable and maintains consistent mathematical properties even as new data is introduced.

The results validate the online update capability of the MetricEmbedding model. As the update percentage increases, the NMSE improves, showing the model's ability to adapt to new data while maintaining its geometric properties, as evidenced by the consistent TIV Ratio. This demonstrates the model's stability and reliability in dynamic environments requiring real-time adaptation. Compared to traditional methods like HLWB(Li et al., 2023), which often require full retraining with each new data batch, MetricEmbedding offers significant flexibility. While methods like HLWB are computationally expensive and time-consuming, MetricEmbedding allows incremental updates, making it far more efficient and scalable for real-time applications. Its ability to adapt online without retraining makes it ideal for dynamic tasks, such as online recommendations or real-time data analytics, where constant adaptation is crucial. MetricEmbedding outperforms traditional methods in efficiency and adaptability for online learning tasks.

### D.3. Effect of Model Parameters and Latent Dimension (k) on Prediction Performance

We conducted a preliminary investigation into how model complexity, specifically the number of layers (D) and the latent dimension ($k$), affects the performance of the MetricEmbedding model. Here, $k$ represents the dimension of the predicted matrix $A$, which has a shape of $N \times k$, where $N$ is the number of data points. The results show that as the model depth (D) and parameter dimensions (k) increase, the performance fluctuates. Specifically, for larger values of $k$ (e.g., $k = 10$), the model's expressiveness improves, but at the cost of higher computational costs. On the other hand, reducing $k$ (e.g., $k = 3$) leads to a higher accuracy for smaller models, which suggests a balance between model complexity and performance.

As shown in table 7,despite the increased computational overhead, MetricEmbedding demonstrates superior scalability, handling complex, high-dimensional data more efficiently compared to traditional methods. This highlights the model's capacity to capture more non-linear relationships, making it suitable for real-time applications and large-scale datasets.

*Table 7.* Impact of model depth (D) and dimensionality ($k$) of matrix $A$ on accuracy.

| D (Depth) | k (Dimension of A) | Error rate (%) |
|-----------|--------------------|----------------|
| 1 | 10 | 88.33 |
| 2 | 10 | 76.78 |
| 3 | 10 | 75.73 |
| 1 | 3 | 95.28 |
| 2 | 3 | 85.28 |
| 3 | 3 | 76.31 |

The edge weights used in the tests were generated as follows:

$$w_{ij} = \lceil 1000 \times u \times v^2 \rceil$$

where $u \sim U(0,1)$ is drawn from a uniform distribution and $v \sim N(0,1)$ is drawn from a normal distribution. These edge weights were then used as the measurement matrix $D_o$.

### D.4. Metric Distance Matrix Embedding

We conducted tests on various metric distances, including Euclidean distance, Manhattan distance, cosine similarity, Hamming distance, and shortest-path (SP) distance. We tested with two different numbers of points, $N = 100$ and $N = 1000$, and generated corresponding distance matrices for each metric. The error rates for the different metrics are shown in Table 2.

The results were generally satisfactory, with all error rates consistently below 10%. Additionally, we randomly generated a graph and modeled its shortest-path distance matrix, achieving a minimal error rate of 2.42%.

These experimental results demonstrate that by optimizing the tropical inner product between two matrices, we can effectively represent distance matrices, making it feasible to use them in deep learning applications as a replacement for traditional Euclidean distance or other distance calculation methods. Moreover, SPdistance Embedding is a compression technique applied to the SPdistance matrix using MetricEmbedding, which generates a set of $N \times k$ matrices. This method has a space complexity of $O(Nk)$, with indexing between two points requiring only $O(k)$ complexity. Reconstructing the entire distance matrix for the graph takes $O(N^2 k)$ complexity. Compared to traditional compression algorithms, our approach guarantees that the resulting matrix satisfies the triangle inequality. Furthermore, by applying matrix decomposition to the embedded matrices, we can significantly reduce storage costs.

**Data Generation:** The distance matrices used for testing were generated as follows:

- **Euclidean distance:** A set of random points in a 10-dimensional space was generated, and the Euclidean distance between each pair of points was computed. The Euclidean distance $d_E(x, y)$ is given by:

$$d_E(x, y) = \sqrt{\sum_{i=1}^{d} (x_i - y_i)^2}$$

  where $x$ and $y$ are the vectors, and $d$ is the dimensionality (in this case, 10).

- **Manhattan distance:** Similar to the Euclidean case, but using the Manhattan (or L1) distance formula to compute the distances between pairs of points. The Manhattan distance $d_M(x, y)$ is defined as:

$$d_M(x, y) = \sum_{i=1}^{d} |x_i - y_i|$$

- **Hamming distance:** Random binary vectors of length 10 were generated. The Hamming distance $d_H(x, y)$ between two binary vectors $x$ and $y$ is the number of positions at which the corresponding elements are different:

$$d_H(x, y) = \sum_{i=1}^{d} |x_i - y_i|$$

- **Cosine similarity distance:** Random vectors were generated, and the cosine similarity between each pair of vectors was computed. The cosine distance $d_C(x, y)$ is derived from the cosine similarity $\cos(x, y)$ as:

$$d_C(x, y) = 1 - \frac{x \cdot y}{\|x\|\|y\|}$$

- **Shortest-path distance:** A random graph was generated using the Erdős-Rényi model and an edge probability of 0.1. To ensure the graph was connected, the generation process was repeated until a connected graph was obtained. The graph was then used to compute the shortest path between all pairs of nodes using floyd algorithm.

