# OpenReview forum: "MetricEmbedding: Accelerate Metric Nearness by Tropical Inner Product"
_ICML.cc/2025/Conference — ICML 2025 poster_

### Official Review · Reviewer_pa67 · 2025-03-10

**Overall Recommendation:** 3

**Summary:**

This paper considers the problem of metric nearness and proposes a new method for solving the problem. In particular, they define a tropical neural network that uses tropical operations instead of regular operations, a new loss function to be used to train the model, and then experimental results validating their model.

**Claims And Evidence:**

The main contributions of the paper are

1. Theorem 8.

I think this claim has sufficient evidence. The proof from skimming seems correct to me.

2. Empirically, testing their new method.

There is quite a bit of evidence for this claim; however, I think there could be more evidence, particularly explicit comparisons against Project and Forget.

**Essential References Not Discussed:**

The paper does a good work of providing references for the $\ell_2$ version of the metric nearness problem. That is when the loss function is the squared distance. However, if the paper chooses to do so, I think it could improve the paper to connect to broader literature for the $\ell_0$ version of the problem. I provide the necessary papers here [A,B,C,D].

Additionally, in relation to the downgrade of the performance when it is not a metrix (Line 19 Right Col). The experiments in the introduction are nice conrete examples [B,E]

There also have been targeted works, where the target is a tree metrix [G] and Euclidean metric [H]. There also have been older work that tried fixing the metric [I]

[A] A. C. Gilbert and L. Jain. If it ain’t broke, don’t fix it: Sparse metric repair. In 2017 55th Annual Allerton
Conference on Communication, Control, and Computing (Allerton), pages 612–619, Oct 2017.

[B] Chenglin Fan, Anna C .Gilbert, Benjamin Raichel, Rishi Sonthalia, and Gregory Van Buskirk. Generalized metric repair on graphs. In Susanne Albers, editor, 17th Scandinavian Symposium and Workshops on Algorithm Theory, SWAT 2020, June 22-24, 2020, Tórshavn, Faroe Islands, volume 162 of LIPIcs, pages 25:1–25:22. Schloss Dagstuhl - Leibniz-Zentrum für Informatik, 2020.

[C] Vincent Cohen-Addad, Chenglin Fan, Euiwoong Lee, and Arnaud de Mesmay. Fitting metrics and ultrametrics with minimum disagreements. In 63rd IEEE Annual Symposium on Foundations of Computer Science (FOCS), pages 301–311. IEEE, 2022

[D] Chenglin Fan, Benjamin Raichel, and Gregory Van Buskirk. Metric Violation Distance: Hardness and Approximation. Proceedings of the Twenty-Ninth Annual ACM-SIAM Symposium on Discrete Algorithms, pages 196–209, 2018.

[E] Anna C. Gilbert and Rishi Sonthalia. Unsupervised metric learning in presence of missing data. 2018 56th Annual Allerton Conference on Communication, Control, and Computing (Allerton), pages 313– 321, 2018.

[G] Vincent Cohen-Addad, Debarati Das, Evangelos Kipouridis, Nikos Parotsidis, and Mikkel Thorup. Fitting distances by tree metrics minimizing the total error within a constant factor. In 62nd IEEE Annual Symposium on Foundations of Computer Science (FOCS), pages 468–479. IEEE, 2021.

[H] Rishi Sonthalia, Greg Van Buskirk, Benjamin Raichel, and Anna C. Gilbert. How can classical mul- tidimensional scaling go wrong? In Advances in Neural Information Processing Systems (NeurIPS), pages 12304–12315, 2021.

[I] Julian Laub,Klaus-Rober tMüller,Felix A Wichmann, and Jakob H Macke. Inducing metric violations in human similarity judgements. In Advances in neural information processing systems, pages 777–784, 2007.

**Experimental Designs Or Analyses:**

The experimental design is valid for section 4.2. Except there is a case where the paper does not test that should be tested. Specifically, metrics where bullet point 3 (Page 3 Line 156 right col) is violated.

For the experiment in section 4.3, error rate is not defined. Hence makes the experiement difficult to understand.

The experiment in section 4.4 does not make sense. How is the new data fed in? Is the original matrix masked and the mask is then slowly changed? Also if $d(a,b)$ is masked. Can $d(a,c)$ be visible to the method earlier. When testing, why should the trained network have any information about the new data?

Section 4.5 is okay

Section 4.6 error rate is again not defined. Additionally, metrix nearness finds the closest metrix. There is no reason to assume that the closest metric is of the same type as the metric that was used to create the corrupted matrix.

**Methods And Evaluation Criteria:**

The paper should benchmark against Project and Forget.

One particular dataset that could be added is the case are metrics where bullet point 3 (Page 3 Line 156 right col) is violated.

**Other Comments Or Suggestions:**

There are some typos such as missing spaces.

**Other Strengths And Weaknesses:**

The main weakness for me are sections 4.3, 4.4 and 4.6. If the authors clarify the concerns there. I am happy to increase my score.

**Questions For Authors:**

Please see the experimental design section. I have a variety of questions about Section 4.3, 4.4 and 4.6

**Relation To Broader Scientific Literature:**

The metric nearness problem is an important problem, and solutions to it help design robust and theoretically justified data analysis techniques.

However, solving the problem is quite computationally intensive. Hence advances in making solving the problem more tractable are important. Additionally, the paper uses neural networks. This makes it more tractable to integrate into the broader framework.

**Theoretical Claims:**

I skimmed the proofs. They look okay.

However, I do not think Theorems 1,2,3,4 need to be presented. These are quite standard.

---

> ### Author Rebuttal · Authors · 2025-04-01
>
> **Q1:Undefined error rate in Sec 4.3**
>
> A1:Thank you for pointing this out. We agree that the evaluation metric in Section 4.3 should be clarified.
>
> All experiments in Section 4.3 use **Normalized Mean Squared Error (NMSE)**, as defined in Section 4.2, to measure reconstruction quality. Specifically, we:
>
> - Compute the ground-truth distance matrix using pairwise Euclidean distances;
> - Apply noise or masking to simulate degradation;
> - Use NMSE to quantify both pre- and post-recovery error;
> - Set missing entries to zero to ensure consistent NMSE calculation.
>
> For example, in the M30 setting, NMSE drops from 35.50% (before recovery) to 13.85% (after), showing a clear performance gain.
>
> Our method leverages the assumption that mildly corrupted metric-consistent matrices retain latent structure, and that enforcing metric constraints during recovery improves reconstruction accuracy.
>
> **Q2:Data handling ambiguity (Sec 4.4)**
>
> A2:Thank you for your comment. Section 4.4 demonstrates the **online update capability** of our method—an aspect largely overlooked in prior work.
>
> We simulate an online setting where the model is incrementally updated as new data arrives, without access to future entries. Starting from a partially observed distance matrix (e.g., 20% revealed), we iteratively expose 20% new entries at each timestep. The model is updated using only these newly observed entries.
>
> This setup respects causality and highlights our method’s ability to **refine predictions progressively**. Traditional convex optimization methods require full data access and incur high computational costs (e.g., \(O(N^3)\)), making them impractical for real-time updates.
>
> In contrast, our method uses **efficient, localized gradient-based updates** via backpropagation, avoiding full retraining. As demonstrated in our experiments, this approach achieves **10×–1000× speedups** over baseline methods while maintaining or improving accuracy. These properties make our method well-suited for **real-time applications** that demand continuous adaptation.
>
> **Q3:Error undefined, metric mismatch (Sec 4.6)**
>
> Thank you for your comment.
>
> Regarding the evaluation metric in Section 4.6, we again adopt the Normalized Mean Squared Error (NMSE) to quantify how well the compressed representation preserves the original distances.
>
> Regarding the assumption about metric nearness, we agree that the closest metric under a given norm is not necessarily of the same type as the one used to generate the original distances (e.g., Euclidean, cosine, etc.). However, the goal of this experiment is not to recover the exact generating metric, but rather to demonstrate that our method can serve as a general-purpose, low-rank, metric-preserving approximation of distance matrices. Specifically A fully specified distance matrix typically requires $O(n^2)$ parameters.  Our method seeks to compress this representation to  O(nk) parameters, while:
>
> - (a) Guaranteeing that the reconstructed matrix satisfies the metric properties, and
>
> - (b) Maintaining low computational complexity, suitable for large-scale applications.
>
> Traditional methods such as SVD or spectral decomposition offer low-rank approximations but do not guarantee metric validity, and
> often incur $ O(n^3)$ complexity. In contrast, our method can be viewed as a tropical analogue of SVD, which:
> - Operates in a multiplication-free space,
> - Enforces metric constraints by design,
> - And achieves efficient approximation using only O(n·k) parameters.
> In this experiment, we evaluate performance in a controlled setting where the original distance matrices are generated using known distance functions .  Our method does not assume knowledge of the underlying metric but still achieves high-fidelity reconstructions, as evidenced by low NMSE and strong performance in downstream tasks (e.g., 0.79 accuracy on the Cora dataset).
>
> **Q4:Uncovered case: bullet 3 violation (Sec 4.2)**
>
> A4：Thank you for your comment. We will revise **Theorem 8** to clarify that satisfying both **Bullet 2** and **Bullet 3** implies the **triangle inequality**.
>
> Our experiments deliberately use matrices with heavily violated metric properties—e.g., **99.7%** of entries in the graph-t1 dataset (\(N=1000\)) violate **Bullet 3**.
>
> Q5: Unnecessary Theorems 1–4
>
> A5: Thank you. We agree that Theorems 1–4 are standard and will streamline or move them to the appendix to improve clarity without losing completeness.
>
> Q6: Essential References Not Discussed
>
> A6: Thank you for the suggestion. We have reviewed references [A–I] and will revise the related sections accordingly. Detailed updates will be provided in the next rebuttal stage.
>
> Q7:PAF
>
> For \(N = 2000\), our method (Tropical) achieves an NMSE of **0.18** in **34s** with **0 violations**, while PAF yields a lower NMSE (**0.068**) but takes **3133.91s**, produces **3.7e7 violations**, and fails to converge—highlighting our method’s superior efficiency, and suitability for large-scale applications.

---

> > ### Comment · Reviewer_pa67 · 2025-04-01
> >
> > Thank you for the clarification.
> >
> > With the clarifications and adding PAF to Table 1, I am willing to increase my score.
> >
> > The authors' comment on PAF taking 3k seconds for N = 2000 implies that it is faster than TRF and HLWB. Also, while 3.7e7 sounds large when dividing by $\binom{2000}{3}$, it is approximately 2.8%. Hence, this has fewer violations than TRF. Hence, PAF is a good baseline and should be included in a complete comparison.

---

> > > ### Author Response · Authors · 2025-04-07
> > >
> > > **Q1:PAF**
> > >
> > > Thank you for the helpful suggestion. We agree that PAF is an important baseline and we will add comprehensive comparisons across datasets, including large-scale scenarios. Our revised experiments address prior omissions and introduce a new noisy MNIST dataset. Metrics include NMSE and convergence time, with convergence defined as stabilization of both NMSE and violation rate.
> > >
> > > **The Tropical method demonstrates several key advantages:**
> > >
> > > - **Efficiency:** Achieves up to 50–60× speedup over PAF on large datasets (N=1000)
> > >
> > > - **Constraint Satisfaction:** Guarantees 0% metric violation across all settings, as ensured by Theorem 6.
> > >
> > > - **Online Updates:** Supports real-time updates, enabled by the short per-epoch computation time.
> > >
> > > In terms of **accuracy**, the slightly higher NMSE compared to PAF is mainly due to the limited optimization space and susceptibility to local minima. This issue can be alleviated through a multi-start strategy that explores more diverse solutions.
> > >
> > > Tropical’s efficiency, constraint guarantees, and scalability make it well-suited for time-sensitive metric nearness tasks. We appreciate your emphasis on PAF and have included thorough comparisons to ensure completeness.
> > >
> > > ### Table 1. Comparison of Methods for Different Matrix Sizes.
> > >
> > > | Matrix Size (N) | Method | Computation Time (s)     | NMSE (Ratio) | Triangle Inequality Violations (%) |
> > > |------------------|---------------------|----------------------------|---------------|-------------------------------------|
> > > | 100              | Ours            | 0.69     | 0.084        | 0%    |
> > > |                  | HLWB                | 14.80     | 0.072        | 0% |
> > > |                  | TRF                 | 12.09         | 0.059        | 4.71%  |
> > > |                  | PAF                 | 21.844                     | 0.071        | 0%   |
> > > | 500              | Ours         | 16.39                      | 0.099         | 0%    |
> > > |                  | HLWB                | 1291.61                    | 0.069         | 0%           |
> > > |                  | TRF                 | 1120.73                    | 0.058         | 4.53%                               |
> > > |                  | PAF                 | 266                        | 0.069        | 0%                                  |
> > > | 1000          | Ours            | 26.73                      | 0.136        | 0%                                  |
> > > |                  | HLWB                | >2000                      | 0.068        | 0%                                  |
> > > |                  | TRF                 | >2000                      | 0.058        | 4.85%                               |
> > > |                  | PAF                 | 1619.68                    | 0.068        | 0%                                  |
> > >
> > > ### Table 2. Experimental Results on Noisy MNIST Distance Matrix
> > >
> > > | Matrix Size (N) | Method     | Computation Time (s) | NMSE (Ratio) | Triangle Inequality Violations (%) |
> > > |------------------|------------|-----------------------|---------------|-------------------------------------|
> > > | 100              | Ours   | 0.53                  | 0.063         | 0%                                  |                             |
> > > |                  | PAF        | 4.91                  | 0.055         | 0.05%         |
> > > | 500              | Ours  | 5.73                  | 0.086         | 0%     |
> > > |                  | PAF        | 55.91                 | 0.055         | 0.18%      |
> > > | 1000             | Ours   | 7.87                  | 0.117         | 0%                                  |
> > > |                  | PAF        | 374.91                | 0.055         | 0.2%                                |
> > >
> > > ### Table 3. Per-Epoch Runtime Comparison (Same Setting as Table 1, N = 500)
> > > | Matrix Size (N) | Method   | Epochs | Total Time (s) | Avg Time per Epoch (s) |
> > > |------------------|----------|--------|------------------|--------------------------|
> > > | 500              | Tropical | 20     | 0.734            | 0.036                    |
> > > |                  | PAF      | 20     | 60.34            | 3.02                     |
> > >
> > > **Q2:Essential References Not Discussed**
> > >
> > > Thank you for your comments. We will expand the *Related Work* section to include discussions on sparse metric repair in Article [A], its graph extension in Article [B], inconsistency minimization in Article [C], "metric violation distance" complexity in Article [D], unsupervised metric learning with missing data in Article [E], tree metric fitting in Article [G], Euclidean metric challenges in Article [H], and early work on metric violations in human similarity judgment in Article [I]. This will provide important context for our work.
> > >
> > > Regarding the $ L_0 $ norm, we cannot apply it due to the lack of gradients. However, we can extend our method to the $ L_1 $ norm and other differentiable norms, preserving sparsity and enabling gradient-based optimization. We plan to explore the $ L_0 $ norm version in future work.

---

### Official Review · Reviewer_MWLk · 2025-03-17

**Overall Recommendation:** 4

**Summary:**

The paper introduces MetricEmbedding, a novel approach using the ropical inner product (max-plus operation) to efficiently solve the Metric Nearness Problem (MNP) while ensuring metric properties like the triangle inequality. The authors showed the equivalence (up to diagonal elements) between the class of non-negative distance matrices and the set of matrices resulting from the tropical inner product of non-negative matrices. Using this observation, the authors proposed  a continuous optimization task to efficiently solve the Metric Nearness Problem (MNP). The proposed method has been shown to significantly reduce computational complexity, achieving up to 1000× speed improvements over traditional approaches while scaling to large matrices (10^5 \times 10^5) with lower memory usage. Experimental results demonstrate its effectiveness in restoring metric properties, handling noisy and incomplete data, on synthetic datasets.

**Claims And Evidence:**

Most claims are clear and convincing, namely, the investigation of the the relationship between tropical operations and metric matrices, presented in Section 3.1 and Appendix A.
The optimization strategy proposed in Section 3.2. potentially requires further theoretical evidence for its ability to achieve plausible minimum for the proposed optimization problem. Specifically, in Algorithm 1 - "Training Procedure for MetricEmbedding", and the accompanying text, the authors propose a two-step parameter update: an RMSProp step followed by changing W_i, b_i to be non-negative. It is unclear whether the proposed approach is guaranteed to achieve a plausible minimum and under which conditions.

**Essential References Not Discussed:**

I cannot comment because the paper is not in my area of expertise.

**Experimental Designs Or Analyses:**

The experiments were performed using synthetic dataset. The experiments look sound and provide extensive evaluation of the proposed method. See my comment about lack of experiments with real world data, in the "Methods And Evaluation Criteria" section.

**Methods And Evaluation Criteria:**

The method was evaluated on synthetic datasets (per Section 4.1.). The metrics used to evaluate the performance make sense for MNP.
Lack of evaluation on non-synthetic datasets makes it unclear how applicable the method is for real world applications.

**Other Comments Or Suggestions:**

See the above.

**Other Strengths And Weaknesses:**

Strengths.
* The paper is very clearly written and easy to follow.
* Most claims have supporting proofs.
* Experiments validate significant improvement achieved by the proposed method over state of the art approaches, with comparable or better accuracy.

**Questions For Authors:**

Please clarify whether the proposed training procedure for MetricEmbedding is guaranteed to achieve a plausible minimum and under which conditions.

**Relation To Broader Scientific Literature:**

I cannot comment because the paper is not in my area of expertise.

**Theoretical Claims:**

I checked the theorems and proofs in Section 3.1 and Appendix A, for correctness.
See the description in the "Claims And Evidence" section above for an open question about convergence guarantees for the proposed optimization algorithm.

---

> ### Author Rebuttal · Authors · 2025-04-01
>
> **Q1：Please clarify whether the proposed training procedure for MetricEmbedding is guaranteed to achieve a plausible minimum and under which conditions.**
>
> A1：The short answer is **no**, because it is a **non-convex** optimization problem. Nevertheless, our algorithm is guaranteed to converge to a local minimum, as we solve the problem using a gradient descent approach.
>
> We prove this by the following:
>
> (1) consider one layer case，
> $
> \min_X \| X \odot_{\max} X^T - Y \|_F^2
> $
> Let:
> $
> A = [[1,3,3],[3,1,3][3,3,1]],
> B = [[3,1,1][1,3,1][1,1,3]]
> $
> Taking \(\alpha = \frac{1}{2}\), then:
> f(\frac{1}{2}A + \frac{1}{2}B) > \frac{1}{2}f(A) + \frac{1}{2}f(B)
> This violates convexity, proving that even in a single-layer case, the problem is **non-convex**.
> (2) Now consider the general multi-layer formulation，We can instead set all parameters of the network to zero except for those in the first layer, keeping the form consistent with the one-layer result described above.
> Then the entire deep network reduces to the form \( W_0 \), and we recover exactly the same objective as in the single-layer case.
>
> **end of proof**
>
> Since global optima are hard to attain in non-convex problems, we instead focus on finding **good local optima**. Our approach achieves a **balance between optimization space and efficiency**.
>
> The original problem is defined under standard metric constraints (triangle inequality and zero diagonals). Our method introduces a stricter condition (**Theorem 8 bullet point 3 **), which defines a narrower solution space we refer to as the **tropical metric space**.
>
> While existing methods like HLWB optimize over the general metric space, our approach is more effective when the optimal solution lies close to this constrained space. However, we currently lack a formal proof of convergence.
>
> To encourage convergence to good local minima, we adopt several practical strategies:
> - Initialize outputs around the mean of the target matrix \(D\), ensuring triangle inequality via a constructed matrix \(M\).
> - Add controlled randomness for better optimization dynamics.
> - Use **parallel optimization** to avoid poor local minima, which significantly improves results.
>
>
> **Q2:Lack of evaluation on non-synthetic datasets makes it unclear how applicable the method is for real world applications.**
>
> A2: Thanks for your feedback. To address your concern about the real-world applicability of our method, we conducted additional experiments on the MNIST dataset and a real graph dataset.
>
> To further validate our method for **metric matrix recovery**, we tested it on MNIST using two non-metric cases: noisy Euclidean distances  following [1] and naturally non-metric cosine similarities following.
> ### Noisy Euclidean Distance (MNIST)
>
> | N   | Method    | Used Time (s) | Result       | Violation |
> |------|-----------|----------------|--------------|-----------|
> | 100  | HLWB      | 35.04          | 0.052        | 0         |
> | 100  | Tropical  | 0.53           | 0.063        | 0         |
> ### Cosine Similarity Distance (MNIST)
>
> | N   | Method    | Used Time (s) | Result        | Violation |
> |------|-----------|----------------|---------------|-----------|
> | 100  | HLWB      | 109.11         | 3.50e-06       | 0         |
> | 100  | Tropical  | 0.51           | 0.004          | 0         |
>
> ## Experiments on a Real Graph Dataset
>
> To further validate the practicality of our approach, we applied **MetricPlug** (Section 3.3) to a real-world graph learning task.
>
> ### Task Setup
> We used graph contrastive learning (based on GRACE [2]) for node classification, replacing standard similarity metrics with **MetricPlug** using the tropical inner product.
>
> ### Dataset and Baselines
> We experimented on the Cora dataset (2,708 nodes), with train/val/test splits following GCN [3]. MetricPlug was compared against cosine, Hamming, Euclidean, and Manhattan distances.
>
> ### Experimental Settings
> We used PyG’s `dropout_adj` to perturb edges (ratio = 0.1) for data augmentation. Training was run for 1,000 epochs with a learning rate of 0.01 and 64-dimensional hidden/projection layers. Accuracy was used for evaluation.
>
> | Method       | Validation Accuracy (%) | Test Accuracy (%) |
> |--------------|-------------------------|-------------------|
> | Cosine       | 77.4                    | 76.5              |
> | Manhattan    | 79.0                    | 79.2              |
> | Euclidean    | 78.2                    | 79.0              |
> | MetricPlug   | **79.2**                 | **79.4**          |
> MetricPlug outperformed other methods in both validation and test accuracy.
> [1] Li W, et al. Metric nearness made practical. AAAI2023
> [2] Zhu, Y. Deep graph contrastive representation learning. arXiv.
> [3] Kipf, T. N. Semi-Supervised Classification with Graph Convolutional Networks. ICLR2016.

---

### Official Review · Reviewer_eADx · 2025-03-19

**Overall Recommendation:** 3

**Summary:**

The authors propose the use of tropical algebra to frame the metric nearest problem within a continuous optimization framework. They first demonstrate that the set of non-negative matrices satisfying the triangle inequality can be fully represented using a combination of tropical algebraic representations. They then propose an MLP based on tropical algebraic operations, which they optimize using RMSprop to solve for the nearest valid metric distance matrix to the original matrix.

## Post-Rebuttal update:
Having reviewed the authors' responses to both my own review as well as that of other reviewers, I retain my original rating and would be happy to recommend this work for acceptance to the main conference.

**Claims And Evidence:**

- Optimizing through an MLP-like structure is preferable to directly optimizing over a single starting matrix and avoids local minima
  - There are certainly scale-related benefits to using the MLP-like structure, specifically the mini-batch setup, but one can also easily optimize over multiple starting matrices A in parallel as another means of avoiding local minima.
- I don't think I saw any evidence supporting the purported downstream task benefits of MetricPlug as proposed in section 3.3, specifically compared to existing approaches.

**Essential References Not Discussed:**

N/A

**Experimental Designs Or Analyses:**

No issues in existing proposed experiments and analysis

**Methods And Evaluation Criteria:**

The proposed evaluation is sufficient for demonstrating the soundness of their technique. However I think what remains to be demonstrated is downstream task impact, as the authors claim existing methods may struggle to accurately capture complex relationships for contrastive learning setups.

**Other Comments Or Suggestions:**

Some of the math notation can be cleaned up a bit.
- For example, it's a bit confusing to have A be set of matrix pairs in theorem 8, while also representing individual matrices in other theorems.
- X in L209 column 2 has not been previously defined
- The font for superscripts and subscripts such as off, max, min, is inconsistent.

**Other Strengths And Weaknesses:**

Strengths:
- Appears to be a theoretically sound and practical approach to the metric nearness problem -- specifically so at larger scales
- Proposed solution is fairly simple to implement
Weaknesses:
- Claims of downstream task impact need to be validated

**Questions For Authors:**

- Can the miniabatch-based algorithm in appendix C.2 guarantee the triangle inequality property at a global scale across all pairs i,j? Or only up to its maximum output shape?

**Relation To Broader Scientific Literature:**

The authors compare against TRF and HLWB, which appear to be the most recent and likely state of the art solutions to the metric nearness problem. Unfortunately I am not intricately familiar with the prior literature on this topic, so I cannot comment on any other prior works.

**Theoretical Claims:**

I went through the theorems but did not carefully validate each proof.

---

> ### Author Rebuttal · Authors · 2025-04-01
>
> **Q1:but one can also easily optimize over multiple starting matrices A in parallel as another means of avoiding local minima.**
>
> A1:Thanks for your valuable suggestion. We conducted experiments using the proposed strategy with a single-layer model, and indeed observed further performance improvements over the baseline approach.
> In our experiments, we constructed a set of non-metric matrices using the MNIST dataset, following the approach described in [1].
> We fixed the number of model iterations to **1000**. Under this configuration:
> The convergence error improved from 0.0583 to **0.0536**.
> The improved model required **3.58 seconds**, compared to **0.97 seconds** for the original method.
> This demonstrates a clear **trade-off between convergence quality and computational cost**.
>
> **Q2：I don't think I saw any evidence supporting the purported downstream task benefits of MetricPlug as proposed in section 3.3, specifically compared to existing approaches.**
>
> A2: We conducted additional experiments on real-world tasks to demonstrate the applicability of our approach, specifically MetricPlug as described in Section 3.3.
>
> - **Task Definition:**
>  We applied graph contrastive learning, an unsupervised method for learning node representations using contrastive loss. Specifically, we used the approach from GRACE [2], which involves perturbing edges to generate augmented views. Unlike traditional methods that use cosine or Euclidean similarity metrics, we replaced them with MetricPlug, which uses the tropical inner product to calculate node pair similarity, satisfying the triangle inequality. The evaluation task focuses on node classification.
>
> - **Dataset and Baseline:**
>   We used the Cora dataset, which contains 2708 nodes, and follows the standard data split used in GCN [3]. We compared our MetricPlug method with existing methods based on cosine, Hamming, Euclidean, and Manhattan distances.
>
> - **Experiment Settings:**
> We utilized the dropout_adj function in PyG to randomly perturb edges with a perturbation ratio of 0.1 to generate augmented views. The configuration used a learning rate of 0.01, 1000 epochs, with hidden and projection dimensions set to 64. Accuracy was the evaluation metric. The results are summarized below:
>
> | Method     | Validation Accuracy (%) | Test Accuracy (%) |
> |------------|-------------------------|-------------------|
> | Cosine     | 77.4                    | 76.5              |
> | Manhattan  | 79.0                    | 79.2              |
> | Euclidean  | 78.2                    | 79.0              |
> | Hamming    | 78.8                    | 79.0              |
> | MetricPlug   | **79.2**                 | **79.4**          |
>
> As shown in the table, the MetricPlug method outperforms other distance-based methods, achieving the best results on both validation and test sets.
>
> **Q3：Can the miniabatch-based algorithm in appendix C.2 guarantee the triangle inequality property at a global scale across all pairs i,j? Or only up to its maximum output shape?
> The answer to the first question is **affirmative**.**
>
> Our proposed minibatch-based training method guarantees that, in any iteration, the model's predictions satisfy the triangle inequality for all index pairs \( (i, j) \). Specifically, as long as the two matrices involved in the final Tropical inner product are constrained to be non-negative, the resulting matrix—obtained via Tropical inner product—will satisfy the triangle inequality for any pair \( (i, j) \).
>
> Although the minibatch algorithm only updates a subset of the model's weights based on the current batch, we explicitly enforce non-negativity constraints on all model parameters throughout training. As a result, regardless of the update stage, the forward pass of the model will always produce a full matrix \( H \) whose entries satisfy the triangle inequality across all index pairs \( (i, j) \).
>
> **For example**, consider a \(1000 \times 1000\) matrix with a batch size of 16. In a single iteration, the model only computes the values for those 16 selected positions and updates the weights based on the corresponding loss. However, since all weights remain non-negative due to the imposed constraints, the full \(1000 \times 1000\) matrix outputted by the model at this point will still satisfy the triangle inequality for any pair \( (i, j) \). This property is guaranteed by **Theorem 6**.
>
> Other cases follow by the same reasoning.
>
> **Q4：Some of the math notation can be cleaned up a bit.**
>
> A4:Thank you very much for pointing out this issue. We will address and correct it in the subsequent version of the manuscript
>
> [1] Li W, et al. Metric nearness made practical. AAAI2023
>
> [2] Zhu, Y et al. Deep graph contrastive representation learning. ArXiv.
>
> [3] Kipf, T. N et al. Semi-Supervised Classification with Graph Convolutional Networks. ICLR2016.

---

### Decision · Program_Chairs · 2025-05-01

**Decision:**

Accept (poster)

**Comment:**

The paper initially received ratings of (3, 3, 2). After the rebuttal, the reviewers updated their ratings to (3, 4, 3), all of which were positive.
According to the reviewers' feedback, the rebuttal effectively addressed their concerns regarding several key areas, such as the evidence supporting the claimed benefits of MetricPlug for downstream tasks, whether the mini-batch algorithm guarantees the triangle inequality property, and the lack of evaluations on non-synthetic datasets.

Since all reviewers expressed positive feedback, the area chair recommends a "weak accept" if there is room in the program. The authors are encouraged to include the discussion and additional experiments in their revision.